# SEEKER: ENHANCING EXCEPTION HANDLING IN CODE WITH A LLM-BASED MULTI-AGENT APPROACH

## ABSTRACT

In real-world software development, improper or missing exception handling can severely impact the robustness and reliability of code. Exception handling mechanisms require developers to detect, capture, and manage exceptions according to high standards, but many developers struggle with these tasks, leading to fragile code. This problem is particularly evident in open-source projects and impacts the overall quality of the software ecosystem. To address this challenge, we explore the use of large language models (LLMs) to improve exception handling in code. Through extensive analysis, we identify three key issues: Insensitive Detection of Fragile Code, Inaccurate Capture of Exception Types, and Distorted Handling Solutions. These problems are widespread across real-world repositories, suggesting that robust exception handling practices are often overlooked or mishandled. In response, we propose *Seeker*, a multi-agent framework inspired by expert developer strategies for exception handling. Seeker uses agents—Scanner, Detector, Predator, Ranker, and Handler—to assist LLMs in detecting, capturing, and resolving exceptions more effectively. Our work is the first systematic study on leveraging LLMs to enhance exception handling practices, providing valuable insights for future improvements in code reliability.

## 1 INTRODUCTION

In the era of large-scale pre-trained code language models (code LLMs) such as DeepSeek-Coder (Guo et al., 2024), Code-Llama (Rozière et al., 2023), and StarCoder (Li et al., 2023b), functional correctness has become the primary method for evaluating these models. For instance, HumanEval (Chen et al., 2021) proposed generating code based on natural language programming problem descriptions and measured performance using the *Pass@k* metric, representing the rate at which generated code passes all test cases within $k$ attempts. Additionally, CoderEval (Yu et al., 2024) and DevEval (Li et al., 2024a) introduced repo-level code generation tasks to evaluate code LLMs in real development scenarios.

As functional correctness improves, research has shifted focus to addressing defects in LLM-generated code. For example, SWE-bench (Jimenez et al., 2024) evaluates LLMs' ability to generate maintenance patches based on real GitHub issues, while SecurityEval (Siddiq & Santos, 2022) assesses the risk of LLMs generating vulnerable code using CWE-defined vulnerabilities. Studies like He & Vechev (2023b) and Li et al. (2024c) explore guiding code generation to avoid common vulnerabilities. Recently, Ren et al. (2023) conducted a study on the performance of LLM-generated code in code robustness represented by exception handling mechanisms, which opened up new explorations for LLM to predict and handle potential risks of generated code itself before a vulnerability occurs.

Despite progress in exception detection and handling techniques, little attention has been paid to standardizing exception mechanisms, especially for custom exceptions and long-tail exception types. We believe that interpretable and generalizable exception handling strategies are crucial yet underestimated attributes in real code development, significantly affecting code robustness and the quality of code LLM training data. This paper explores these neglected aspects and raises the research question: *Do we need to enhance the standardization, interpretability, and generalizability*

---

*Equal contribution.
†Equal Advising.

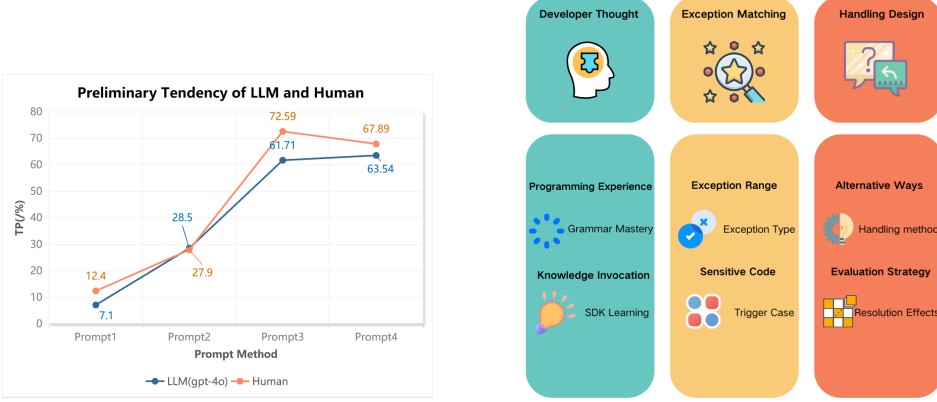

(a) Our preliminary tendency.

(b) a schematic diagram of human developers who well-performed in exception handling.

Figure 1: Preliminary on exception handling performance by LLM and human. Prompt1, Prompt2, Prompt3 and Prompt4 in (a) indicate General prompting, Coarse-grained Knowledge-driven prompting, Fine-grained Knowledge-driven prompting and Fine-grained Knowledge-driven with handling logic prompting respectively

*of exception handling in real code development scenarios?* To the best of our knowledge, no prior work has studied this issue.

To investigate the role of interpretability and rule generalization in exception handling for both human developers and LLMs, we expanded upon preliminary experiments by Ren et al. (2023). We introduced four sets of prompts—Coarse-grained Reminding, Fine-grained Reminding, Fine-grained Inspiring, and Fine-grained Guiding—based on 100 fragile Java code snippets from real projects. These prompts progressively added interpretability and rule generalization to influence code writers' in-context learning. Our findings indicate that code generated with the Fine-grained Guiding prompt exhibits significantly better exception handling performance, while lacking interpretability or rule generalization reduces performance, as shown in Figure 1(a).

Figure 1(b) illustrates the Chain-of-Thought used by senior developers under the Fine-grained Guiding prompt. Notably, rare exceptions like *BrokenBarrierException* and *AccessControlException* can cause high risks but are often poorly handled. Good exception handling practices focus on the specificity of exceptions, accurately capturing exception types deeper in the class hierarchy. Capturing specific exceptions, such as *SQLClientInfoException* over its superclass *SQLException*, provides more detailed error information Osman et al. (2017). However, accurately achieving this remains challenging due to lack of handling paradigms for long-tail or customized exceptions, complex inheritance relationships, and multiple exception handling patterns.

To improve code robustness by leveraging best exception handling practices, we propose **Seeker**, which decomposes exception handling into five tasks handled by specialized agents: **S**canner, **D**etector, **Pr**edator, **Ran**ker, and **Handler**. We build a Common Exception Enumeration (CEE) from trusted external documents to enhance detection, capture, and handling tasks where LLMs perform poorly. This method integrates easily with existing code LLMs to generate highly robust code, and CEE offers community contribution value by helping developers understand ideal exception practices.

However, using Java exceptions as an example, the inheritance tree contains 433 nodes, 62 branches, and 5 layers, making direct retrieval inefficient. To address this, we propose a deep retrieval-augmented generation (Deep-RAG) algorithm tailored for complex inheritance relationships. By assigning development scenario labels to branches and using few-sample verification to fine-tune labels, we improve retrieval performance and reduce overhead. Experiments show that Seeker helps LLMs optimize or generate highly robust code, enhancing performance in various code tasks.

In summary, our main contributions are:

- We highlight the importance of standardization, interpretability, and generalizability in exception handling mechanisms, identifying a gap in existing research.

- We propose **Seeker**, which decomposes exception handling into specialized tasks and incorporates Common Exception Enumeration (CEE) to enhance performance.

- We introduce a deep retrieval-augmented generation (Deep-RAG) algorithm tailored for complex inheritance relationships, improving retrieval efficiency.

- We conduct extensive experiments demonstrating that Seeker improves code robustness and exception handling performance in LLM-generated code.

## 2 PRELIMINARY

### 2.1 MITIGATION EFFECT

In this section, we study how the standardization, interpretability, and generalizability of exceptions affect the exception handling performance of code developers and determine the mitigation effect of poor exception handling. To achieve this, we conduct extensive comparative experiments by controlling the standardization of exception types, the interpretability of risk scenarios, and the generalization of handling strategies, respectively, applying the four sets of in-context learning prompt proposed in figure 4 and 5 (i.e., Coarse-grained Reminding prompting, Fine-grained Reminding prompting, Fine-grained Inspiring prompting, and Fine-grained Guiding prompting).

Specifically, based on the preliminary exploration of Ren et al. (2023), we screened several well-maintained codebases, combined manual and automatic code reviews to filter out high-quality also important exception handling therefore obtain the fragile code that is in serious situation in real development scenarios. Then we allowed code developers to familiarize with these filtered codebases and record the methods and processes they used when handling exceptions. In order to reduce the difficulty of the entire task and simulate the developer's thought about exception handling during the development process, we set up four prompt links to provide developers with progressive exception handling information. The implementation results can be found in figure 1(a).

The comparative experiment reveals an interesting phenomenon: prompts without effective guidance information are not helpful for both human developers and LLMs, while adding type normative information about exception mechanisms will slightly improve developers' vague perception of the source of code fragility, but cannot accurately locate and handle them due to the unfamiliarity with the exception, which is easy to cause insensitive detection. Increasing the interpretability information of the development scenario will greatly improve developers' understanding of the code itself and potential fragility, which is beneficial to the accuracy of exception capture. Increasing the generalization information of handling strategies further improves developers' ability to analyze the source of fragility and improve the quality of handling block. The phenomenon that the above information bring significant gains in exception handling tasks is called the mitigation effect. This phenomenon answers the research questions raised in Section 1 by revealing the mitigation effect by specific prompt information, impacting the quality of code developers' exception handling practices. It also inspires the proposed *Seeker* method to combine external document information to align the generated prompts with fine-grained guidance standards. In addition, Section 3.1 provides a reasonable explanation for the occurrence of the mitigation effect, providing data and insights on the effectiveness of the proposed method. We believe that our findings can provide valuable insights for future research related with reliable code generation, laying the foundation for potential RAG code agent progress.

### 2.2 A REVISIT OF HUMAN EMPIRICALS

Over the years, there have been numerous empirical studies and practical discussions on exception handling, but what is common is that exception handling has been repeatedly emphasized as an important mechanism directly related to code robustness. Nakshatri et al. (2016) points out that exception handling is a necessary and powerful mechanism to distinguish error handling code from normal code, so that the software can do its best to run in a normal state. Weimer & Necula (2004) points out that the exception mechanism ensures that unexpected errors do not damage the stability or security of the system, prevents resource leakage, ensures data integrity, and ensures that the

program still runs correctly when unforeseen errors occur. In addition, Jacobs & Piessens (2009) points out that exception handling also involves solving potential errors in the program flow, which can mitigate or eliminate defects that may cause program failure or unpredictable behavior.

Although the exception mechanism is an important solution to code robustness, developers have always shown difficulties in dealing with it due to its complex inheritance relationship and processing methods. de Pádua & Shang (2017) points out that various programming language projects show a long-tail distribution of exception types when facing exception handling, which means that developers may only have a simple understanding of the frequently occurring exception types. However, according to section1, good exception practices rely on developers to perform fine-grained specific capturing. Nguyen et al. (2020b) also points out multi-pattern effect of exception handling. For example, even for peer code, capturing different exception types will play different maintenance functions, so exception handling is often not generalized or single-mapped. These complex exception mechanism practice skills have high requirements for developers' programming literacy. de Sousa et al. (2020) manually reviewed and counted the exception handling of a large number of open source projects, and believed that up to 62.91% of the exception handling blocks have violations such as capturing general exceptions and destructive wrapping. This seriously violates the starting point of the exception mechanism. de Pádua & Shang (2017) emphasizes the urgent need and importance of automated exception handling suggestion tools.

The failure of human developers in the exception handling mechanism seriously affects the quality of LLM's code training data (He & Vechev (2023a)), which further leads to LLM's inability to understand the usage skills of maintenance functions (Wang et al. (2024)). To solve the above problems, we first proposed $Seeker - Java$ for the Java language. This is because the Java language has a more urgent need for exception handling and is completely mapped to the robustness of Java programs. Ebert et al. (2020) pointed out that as a fully object-oriented language, Java's exception handling is more complex than other languages, and it has a higher degree of integration into language structures. Therefore, Java projects are more seriously troubled by exception handling bugs. In addition, Java relies heavily on exceptions as a mechanism for handling exceptional events. In contrast, other languages may use different methods or have less strict exception handling mechanisms. It is worth mentioning that $Seeker$'s collaborative solution based on an inherent multi-agent framework plus an external knowledge base, they can quickly migrate multiple languages by maintaining documents for different languages. We will also maintain $Seeker - Python$ and $Seeker - C\#$ in the future to provide robustness guarantees for the development of more programming languages.

## 3 METHODOLOGY

In this section, we introduce the proposed **Seeker** method. We first review the historical observations of developers on exception handling issues, and then introduce three exception handling pitfalls, *Insensitive-Detection of Fragile Code*, *Inaccurate-Capture of Exception Type* and *Distorted-Solution of Handling Block*. Finally, we introduce the method's dependency construction and the entire method.

### 3.1 RULES OF GOOD PRACTICE

In this section, we introduce four prompt settings: *Coarse-grained Reminding prompting*, *Fine-grained Reminding prompting*, *Fine-grained Inspiring prompting* and *Fine-grained Guiding prompting*, which can be used to demonstrate the mitigation effect of bad practices on developers when facing exception handling tasks. For *Coarse-grained Reminding prompting*, we use "pay attention to potential exceptions" to remind developers of the exception mechanism, and let developers find the fragile parts of the target code slice and handle them according to their own practical experience. As shown in figure 1(a) , figure 4 and figure 5, although developers will consciously start screening for exception handling, given the difficulties mentioned in Section 2.2, both humans and LLM developers are very insensitive to identifying fragile code. Ren et al. (2023) also found this phenomenon and summarized this series of bad practices as Incorrect exception handling. For *Fine-grained Reminding prompting*, we provide developers with fine-grained reminders of specific exception types based on the fragile code scenario, and let developers understand the source of code fragility and handle it in a standardized manner based on the exception. Although developers will consciously learn from external documents or examples, the information in these documents is often too abstract to

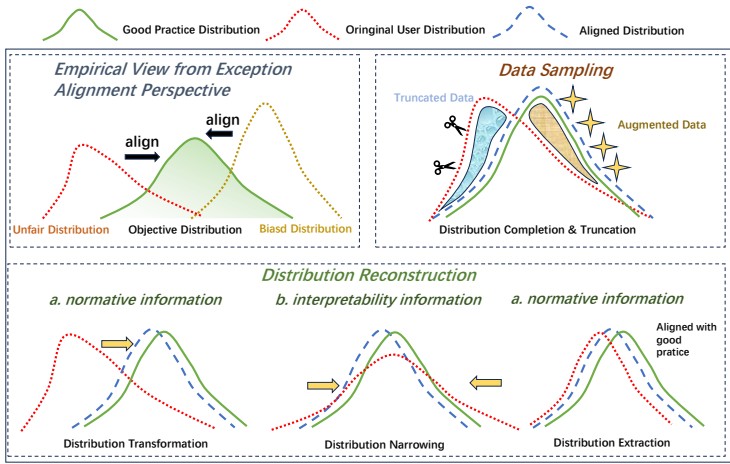

Figure 2: Distribution of Exception Type. Human practice may be far from good practice, thus we conduct data and info processing to align user distribution to good practice.

be interpreted, and as for the examples, most of the time there is no standardized quality assurance or generalization. Therefore, developers tend to catch exceptions inaccurately, and do not fundamentally solve the potential risks of the program. Related studies have shown that the bad practice of Abuse of try-catch often appears in this experimental benchmark. For *Fine-grained Inspiring prompting*, we additionally provide a code-level scenario analysis of the fragile code. Although developers still rely on their own understanding of the code, the intuitive and interpretable natural language significantly improves developers' insight and analysis capabilities for exceptions in this scenario. Related studies also show that for standalone function-level fragile code optimization, this experimental settings can achieve relatively stable good exception handling practices. However, in the face of real development scenarios with complex dependencies, how to generate high-quality handling blocks with generalization is still a challenge. Zhang et al. (2023) pointed out that exception handling code is prone to errors in real projects. For *Fine-grained Guiding prompting*, we additionally give a generalized handling strategy for the exception. Based on the stable exception detection performance of the above experimental benchmarks, developers finally achieve high-quality exception handling practices. de Pádua & Shang (2017) also strongly recommended that developers should use generalizable exception handling strategies, because it is difficult for developers to perform higher-quality optimization before fully mastering the information of an exception type. In essence, these four prompt settings can be regarded as information progression for exception type standardization, fragile interpretability, and handling generalization, thereby changing the developer's in-context learning. By changing the prompts, the robustness of the code generated by the developer will be affected, thereby affecting the quality of the final project. Note that the four sets of prompt we proposed can be applied to any code-based in-context learning, thereby promoting research on the impact of prompt specifications on LLM code generation performance.

Note that for most programming languages, there are three ways to handle exceptions. Exceptions thrown using throws keyword in the method signature, Exceptions thrown using throw keyword in the method body, and Exceptions caught in a try-catch block of a method. Nakshatri et al. (2016) points out that the first method may not provide the real situation, because the exceptions thrown using throws in the method signature will be incorrectly added to the method's call stack, thereby propagating the exception until it is caught. In addition, the exceptions thrown using the second method will eventually be caught by the caller using a try catch block. Therefore, the third method is the most efficient and common exception practice. In our method, we only take the third exception handling way as the best practice when optimize the target.

## 3.2 THE RAG-AGENT METHOD

To enhance the standardization, interpretability, and generalizability of exception handling in real code development scenarios, we propose a method called *Seeker*. Seeker disassembles the chain-of-thought processes of senior human developers and divides the exception mechanism into five specialized tasks, each handled by a dedicated agent: *Planner*, *Detector*, *Predator*, *Ranker*,

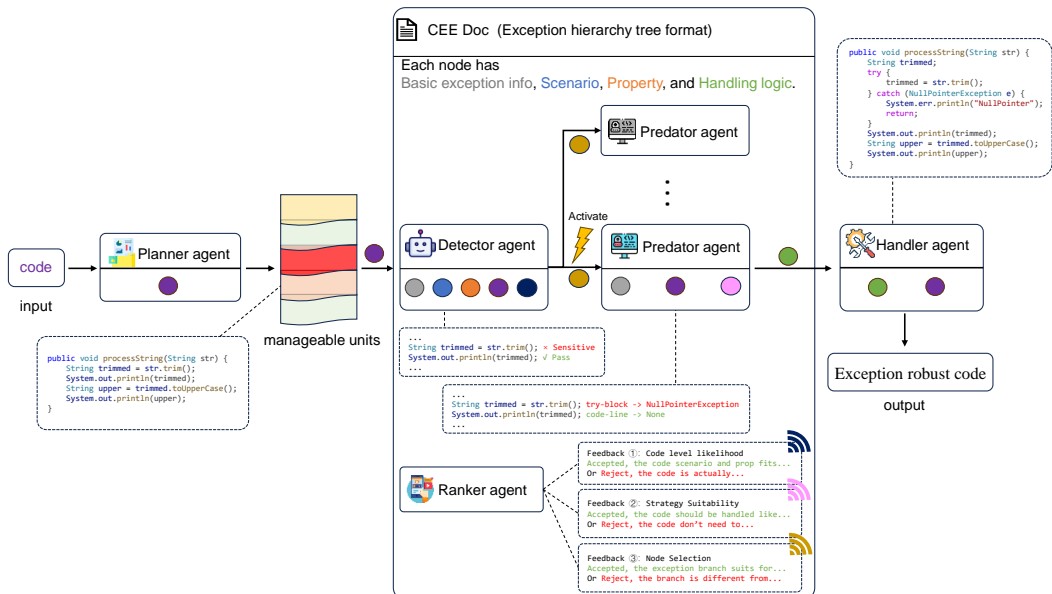

Figure 3: Seeker Work Flow. The workflow consists of four agents: Planner, Detector, Ranker, and Handler, collaborating to manage exception handling in code. The color circle indicates the info passing along the pipeline or used by agents.

and $Handler$. By integrating a large amount of trusted external experience documents with exception practices, we build the $Common\ Exception\ Enumeration\ (CEE)$. CEE is a comprehensive and standardized document providing a structured and exhaustive repository of exception information, encompassing scenarios, properties, and recommended handling strategies for each exception type. The foundation of CEE is detailed in AppendixA.1.2. With the help of CEE, Seeker retrieves and enhances the detection, capture, and handling tasks where the original LLM performs poorly. This method can be easily integrated into existing code LLMs to generate highly robust code, and CEE has promising community contribution and maintenance value, helping developers further understand the ideal practices of exception mechanisms.

Generally, given a piece of code, we first use a planner agent to segment it into manageable units such as function blocks, class blocks, and file blocks. The planner employs a thoughtful approach to segmentation by considering factors such as the overall code volume, dependency levels, and requirement relationships. This strategy helps mitigate the pressure on processing, particularly regarding context window limitations and complex dependency chains, ensuring that no single unit overwhelms the analysis agents. By balancing the granularity of segmentation, we can avoid overly fine divisions that may introduce high complexity, thus maintaining clarity and efficiency in handling large and intricate codebases.

For the $Detector$ agent, it simultaneously performs scenario and property matching alongside static analysis to identify fragile areas in the code that are likely to lead to errors or crashes. These two approaches run in parallel, each contributing their strengths to the detection process. Scenario and property matching offers shallow-level analysis, capturing vulnerabilities based on semantic cues and contextual scenarios that static analysis might overlook due to its challenges in achieving high coverage for exception handling issues. Conversely, static analysis excels in uncovering complex dependencies and deep-level defects, providing insights that shallow analysis may miss. By combining the results from both methods—taking their union—the $Detector$ agent covers both shallow and deep-level risks, effectively detecting potential exceptions with equal consideration for long-tail, domain-specific, or customized exception types. However, as discussed in section 1, detecting exceptions without considering the complex inheritance relationships between exception types may not yield optimal results, as it could lead to inaccurate exception specificity in the exception hierarchy.

Therefore, it is necessary to incorporate external knowledge to guide the capture and analysis processes. To achieve this, we integrate the CEE into the $Predator$ agent. Similar to Retrieval-Augmented Generation (RAG) models, the $Predator$ agent summarizes the code at the function

---

**Algorithm 1:** Seeker Framework

---

**Input:** Codebase $C$
**Output:** Optimized code $C'$ with robust exception handling

1 Segment the codebase $C$ into manageable units $U = \{u_1, u_2, \ldots, u_N\}$;
2 **foreach** *code segment $u_i$ in $C$* **do**
3    **if** *(length of $u_i$ is within predefined limit)* **and** *(function nesting level is low)* **and** *(logical flow is clear)* **then**
4        Add $u_i$ to $U$;

5 Initialize optimized units $U' = \{\}$;
6 **foreach** *unit $u_i$ in $U$* **do**
   `// Detection Phase`
7    Initialize potential exception set $E_i = \{\}$;
8    Use the **Detector** agent to analyze unit $u_i$;
9    **In parallel do** { `// Static Analysis`
10    Generate control flow graph $CFG_i$ and exception propagation graph $EPG_i$ for $u_i$;
11    Identify sensitive code segments $S_i^{\text{static}} = \{s_{i1}^{\text{static}}, s_{i2}^{\text{static}}, \ldots\}$ in $u_i$;
   `// Scenario and Property Matching`
12    Perform scenario and property matching on $u_i$;
13    Identify sensitive code segments $S_i^{\text{match}} = \{s_{i1}^{\text{match}}, s_{i2}^{\text{match}}, \ldots\}$ in $u_i$;
14    } Combine sensitive code segments: $S_i = S_i^{\text{static}} \cup S_i^{\text{match}}$;
15    **foreach** *segment $s_{ij}$ in $S_i$* **do**
16        Detect potential exception branches $E_{bij}$ in $s_{ij}$;
17        $E_{bi} \leftarrow E_{bi} \cup E_{bij}$;
   `// Retrieval Phase`
18    Use the **Predator** agent to retrieve fragile code and try-catch blocks;
19    Summarize unit $u_i$ at the function level to obtain code summary $F_i$;
20    Perform Deep-RAG using $F_i$ and exception branches $E_{bi}$, get exception nodes $E_{ni}$;
21    Mapping relevant exception handling strategies $H_i = \{h_{i1}, h_{i2}, \ldots\}$ from CEE;
   `// Ranking Phase`
22    Use the **Ranker** agent to assign grades to exceptions in $E_{ni}$;
23    **foreach** *exception $e_{ik}$ in $E_{ni}$* **do**
24        Calculate exception likelihood score $l_{ik}$ based on $e_{ik}$ attribute and impact;
25        Calculate suitability score $u_{ik}$ of handling strategy $h_{ik}$;
26        Compute overall grade $g_{ik} = \alpha \cdot l_{ik} + \beta \cdot u_{ik}$;
27    Rank exceptions in $E_{ni}$ based on grades $g_{ik}$ in descending order to get ranked list $E'_{ni}$;
   `// Handling Phase`
28    Use the **Handler** agent to generate optimized code $u'_i$;
29    **foreach** *exception $e_{ik}$ of $E'_{ni}$ if $g_{ik} > \gamma$* **do**
30        Mapping handling strategy $h_{ik}$ from $H_i$;
31        Apply $h_{ik}$ to code segment(s) related to $e_{ik}$ in $u_i$;
32    $U' \leftarrow U' \cup \{u'_i\}$;
33 Combine optimized units $U'$ to produce the final optimized code $C'$;

---

level and queries the CEE for relevant exception attributes. It performs multi-layered deep searches to retrieve information that can be applied to the detected issues, providing valuable context for exception handling. Crucially, during few-shot testing phases, the environment supplies feedback on both the accuracy and coverage of the retrieved information. This feedback is integral to the agent's learning process, enabling it to refine its search strategies and improve the relevance of the information it retrieves. We propose a **Deep Retrieval-Augmented Generation (Deep-RAG)** algorithm to handle the complex inheritance relationships in exception types as further detailed in Appendix A.1.1.

By combining the outputs from the *Detector* and *Predator* agents, the *Ranker* assigns grades to the detected exceptions based on their likelihood and the suitability of the handling strategies retrieved from the CEE. This grading system ensures that *Seeker* prioritizes the most critical ex-

ceptions for immediate handling. The $Ranker$ considers factors such as the likelihood of the exception occurring, the potential impact on the program, and the specificity of the exception type within the inheritance hierarchy. It gives feedback to $Detector$ and $Predator$ agents along with the node selection steps through score ranking and judge, ensuring the agents learning from the actual code environment.

Analyzing the ranked exceptions, the $Handler$ agent generates optimized code that incorporates robust handling strategies. It utilizes templates and logic patterns derived from the CEE to ensure that the generated code is functionally correct. The Handler focuses on capturing accurate fine-grained exceptions, moving down the class hierarchy to provide additional information about errors, beyond what the superclass exceptions provide. This approach helps developers quickly identify the source of the problem, effectively improve the readability and maintainability of the code, and avoid mishandling different types of errors.

However, integrating such a comprehensive exception handling mechanism introduces challenges in computational overhead, especially when dealing with a large number of exception types and complex inheritance relationships. To address this, we designed a high-concurrency interface that keeps the additional computing time overhead constant, regardless of the code volume level. This ensures that the method is scalable and the complexity is controllable when facing any codebase size. We discuss the time costs of Seeker in detail in Appendix A.2.3.

## 4 EXPERIMENTS

In this section, we evaluate the performance of our proposed method, **Seeker**, on the task of exception handling code generation. We aim to answer the following research questions (RQs):

- **RQ1**: How does **Seeker** perform compared to state-of-the-art methods on exception handling code generation tasks?

- **RQ2**: What is the effect of different agents in the **Seeker** framework on the overall performance?

- **RQ3**: How does **Seeker** perform across different evaluation metrics, specifically in terms of code quality and correctness?

- **RQ4**: How does the choice of underlying language model (LLM) affect the performance of **Seeker**?

- **RQ5**: What is the impact of integrating domain-specific knowledge, such as the Common Exception Enumeration (CEE), into **Seeker**?

### 4.1 EXPERIMENT SETUP

#### 4.1.1 DATASETS

We conduct experiments on a dataset consisting of 750 fragile Java code snippets extracted from real-world projects. These code snippets are selected based on their potential for exception handling improvements, following the rules outlined in Appendix A.2.1.

#### 4.1.2 BASELINES

We compare **Seeker** with the following methods:

- **General Prompting**: A straightforward approach where the LLM is prompted to generate exception handling code without any specialized framework or additional knowledge.

- **Traditional Retrieval-Augmented Generation (RAG)**: A method that retrieves relevant information from external sources to assist in code generation.

- **KPC** (Ren et al., 2023): The state-of-the-art method for exception handling code generation, which leverages knowledge graphs and pattern mining.

- **FuzzyCatch** (Nguyen et al., 2020a): A tool for recommending exception handling code for Android Studio based on fuzzy logic.

- **Nexgen** (Zhang et al., 2020): A neural network approach for automated exception handling in Java, which predicts try block locations and generates complete catch blocks in relatively high accuracy.

### 4.1.3 EVALUATION METRICS

To comprehensively assess the effectiveness of our method, we employ six quantitative metrics:

1. **Automated Code Review Score (ACRS)**: This metric evaluates the overall quality of the generated code in terms of adherence to coding standards and best practices, based on an automated code review model.

$$\text{ACRS} = \frac{\sum_{i=1}^{N} w_i s_i}{\sum_{i=1}^{N} w_i} \times 100\% \tag{1}$$

   where:

   - $N$ is the total number of code quality checks performed by the automated code review tool.
   - $w_i$ is the weight assigned to the $i$-th code quality rule, reflecting its importance.
   - $s_i$ is the score for the $i$-th rule, defined as:

$$s_i = \begin{cases} 1, & \text{if the generated code complies with the } i\text{-th rule} \\ 0, & \text{if it does not comply} \end{cases} \tag{2}$$

   A higher ACRS indicates better adherence to coding standards and best practices.

2. **Coverage (COV)**: This metric measures the proportion of actual sensitive code segments that our method successfully detects.

   Let $S = \{s_1, s_2, \ldots, s_N\}$ be the set of actual sensitive code segments.

   Let $D = \{d_1, d_2, \ldots, d_M\}$ be the set of detected sensitive code segments.

   Define an indicator function:

$$I_{\text{detected}}(s_i) = \begin{cases} 1, & \text{if } \exists d_j \in D \text{ such that } d_j = s_i \\ 0, & \text{otherwise} \end{cases}$$

   Then, the Coverage is defined as:

$$\text{COV} = \frac{\sum_{i=1}^{N} I_{\text{detected}}(s_i)}{N} \times 100\%$$

   This metric reflects the percentage of actual sensitive code segments correctly detected by our method. Over-detection (detecting more code segments than actual sensitive code) is not penalized in this metric.

3. **Coverage Pass (COV-P)**: This metric assesses the accuracy of the try-blocks detected by the **Predator** agent compared to the actual code that requires try-catch blocks, penalizing over-detection.

   Let $T = \{t_1, t_2, \ldots, t_P\}$ be the set of actual code regions that should be enclosed in try-catch blocks (actual try-blocks).

   Let $\hat{T} = \{\hat{t}_1, \hat{t}_2, \ldots, \hat{t}_Q\}$ be the set of code regions detected by the **Predator** agent as requiring try-catch blocks (detected try-blocks).

   Define an indicator function:

$$I_{\text{correct}}(\hat{t}_j) = \begin{cases} 1, & \text{if } \hat{t}_j \in T \\ 0, & \text{otherwise} \end{cases}$$

   The number of correctly detected try-blocks is:

$$\text{TP} = \sum_{j=1}^{Q} I_{\text{correct}}(\hat{t}_j)$$

The number of false positives (incorrectly detected try-blocks) is:

$$\text{FP} = Q - \text{TP}$$

The number of false negatives (actual try-blocks not detected) is:

$$\text{FN} = P - \text{TP}$$

We define the Coverage Pass (COV-P) as:

$$\text{COV-P} = \frac{\text{TP}}{P + \text{FP}} \times 100\%$$

This formulation penalizes over-detection by including the false positives in the denominator. A try-block is considered correct if it exactly matches the actual code lines; any over-marking or under-marking is counted as incorrect.

4. **Accuracy (ACC)**: This metric evaluates the correctness of the exception types identified by the **Predator** agent compared to the actual exception types.

   Let $E = \{e_1, e_2, \ldots, e_R\}$ be the set of actual exception types that should be handled.

   Let $\hat{E} = \{\hat{e}_1, \hat{e}_2, \ldots, \hat{e}_S\}$ be the set of exception types identified by the **Predator** agent.

   Define an indicator function:

$$I_{\text{correct}}(\hat{e}_j) = \begin{cases} 1, & \text{if } \exists e_i \in E \text{ such that } \hat{e}_j = e_i \text{ or } \hat{e}_j \text{ is a subclass of } e_i \\ 0, & \text{otherwise} \end{cases}$$

   Then, the Accuracy is defined as:

$$\text{ACC} = \frac{\sum_{j=1}^{S} I_{\text{correct}}(\hat{e}_j)}{S} \times 100\%$$

   This metric reflects the proportion of identified exception types that are correct, considering subclass relationships. Over-detection of incorrect exception types decreases the accuracy.

5. **Edit Similarity (ES)**: This metric computes the text similarity between the generated try-catch blocks and the actual try-catch blocks.

   Let $G$ be the generated try-catch code, and $A$ be the actual try-catch code.

   The Edit Similarity is defined as:

$$\text{ES} = 1 - \frac{\text{LevenshteinDistance}(G, A)}{\max(|G|, |A|)}$$

   where $\text{LevenshteinDistance}(G, A)$ is the minimum number of single-character edits (insertions, deletions, or substitutions) required to change $G$ into $A$, and $|G|, |A|$ are the lengths of $G$ and $A$, respectively.

   A higher ES indicates that the generated code closely matches the actual code.

6. **Code Review Score (CRS)**: This metric involves submitting the generated try-catch blocks to an LLM-based code reviewer (e.g., GPT-4o) for evaluation. The language model provides a binary assessment: *good* or *bad*.

   Let $N_{\text{good}}$ be the number of generated try-catch blocks evaluated as *good*, and $N_{\text{total}}$ be the total number of try-catch blocks evaluated.

   The Code Review Score is defined as:

$$\text{CRS} = \frac{N_{\text{good}}}{N_{\text{total}}} \times 100\%$$

   This metric reflects the proportion of generated exception handling implementations that are considered good according to engineering best practices.

## 4.2 RQ1: PERFORMANCE COMPARISON WITH BASELINES

We compare the performance of **Seeker** with the baselines on the exception handling code generation task. The results are presented in Table 1.

Table 1: Comparison of Exception Handling Code Generation Methods

| Method | ACRS | COV (%) | COV-P (%) | ACC (%) | ES | CRS (%) |
|---|---|---|---|---|---|---|
| General Prompting | 0.21 | 13 | 9 | 8 | 0.15 | 24 |
| Traditional RAG | 0.35 | 35 | 31 | 29 | 0.24 | 31 |
| KPC (Ren et al., 2023) | 0.26 | 14 | 11 | 8 | 0.17 | 27 |
| FuzzyCatch (Nguyen et al., 2020a) | 0.76 | 83 | 77 | 75 | 0.71 | 73 |
| Nexgen (Zhang et al., 2020) | 0.73 | 79 | 74 | 75 | 0.68 | 72 |
| **Seeker (Ours)** | **0.85** | **91** | **81** | **79** | **0.64** | **92** |

As shown in Table 1, **Seeker** outperforms all baselines across all evaluation metrics. Specifically, we observe:

- A significantly higher **ACRS**, indicating superior overall code quality.
- Substantially greater **Coverage (COV)** and **Coverage Pass (COV-P)**, demonstrating **Seeker**'s effectiveness in detecting and correctly wrapping sensitive code regions.
- Higher **Accuracy (ACC)** in identifying the correct exception types, including recognizing subclass relationships.
- An improved **Edit Similarity (ES)**, showing that the generated code closely matches the actual exception handling code.
- A higher **Code Review Score (CRS)**, confirming that our implementations are more frequently deemed good by the LLM reviewer.

These results demonstrate that **Seeker** achieves state-of-the-art performance in exception handling code generation.

In our experiments, we also evaluated the stability of our method against multiple baselines across two key dimensions: the creation time of test code snippets and the function count within these snippets. Performance over time, as shown in Figure 77, indicates that while baseline methods tend to exhibit variability, particularly in recent years, our method consistently maintains high performance levels. This stability suggests that our approach is less sensitive to temporal changes in the test code's development environment, highlighting its robustness in adapting to evolving software trends and requirements.

Similarly, when analyzing performance as a function of code complexity, represented by the number of functions per test snippet, our method demonstrates a clear advantage in stability. Baseline methods generally perform well under simpler conditions (lower function counts) but show significant declines as the complexity of the code increases. In contrast, our method sustains its performance across all levels of code complexity, demonstrating adaptability to more complex test scenarios. This robustness in handling both temporal and complexity-based variations underscores the resilience of our approach, making it a reliable choice in dynamic and evolving code testing environments.

## 4.3 RQ2: EFFECT OF DIFFERENT AGENTS IN **SEEKER**

To understand the contribution of each agent in the **Seeker** framework, we conduct an ablation study by removing one agent at a time. The results are presented in Table 2.

From Table 2, we can observe that removing any agent from the framework leads to a degradation in performance across all metrics. This highlights the importance of each agent's role:

- The **Scanner** agent is crucial for initial code analysis, contributing to all metrics.
- The **Detector** agent enhances the identification of sensitive code regions, mainly affecting COV-P and COV.

Table 2: Ablation Study on the Effect of Different Agents

| Configuration | ACRS | COV (%) | COV-P (%) | ACC (%) | ES | CRS (%) |
|---|---|---|---|---|---|---|
| **Seeker (Full)** | **0.85** | **91** | **81** | **79** | **0.64** | **92** |
| Without Scanner Agent | 0.78 | 85 | 75 | 73 | 0.59 | 86 |
| Without Detector Agent | 0.76 | 63 | 54 | 61 | 0.51 | 84 |
| Without Predator Agent | 0.72 | 61 | 53 | 42 | 0.47 | 81 |
| Without Ranker Agent | 0.63 | 90 | 79 | 75 | 0.49 | 65 |
| Without Handler Agent | 0.50 | 91 | 81 | 79 | 0.34 | 42 |

- The **Predator** agent is key for accurately detecting exception types, mainly impacting ACC, COV and COV-P.
- The **Ranker** agent improves the selection of the best handling strategies, contributing to overall code quality, mainly affecting ES and CRS.
- The **Handler** agent ensures proper implementation of exception handling, affecting ES and CRS.

## 4.4 RQ3: Performance Across Different Metrics

We further analyze **Seeker**'s performance across the different evaluation metrics. The high scores in ACRS and CRS indicate that **Seeker** not only generates code that adheres to best practices but also produces high-quality code as per automated and LLM-based code reviews. The high COV and COV-P scores show that our method effectively detects and correctly wraps sensitive code regions. The high ACC and ES scores demonstrate accurate exception type identification and code similarity to actual implementations.

## 4.5 RQ4: Effect of Underlying Language Model

To evaluate the impact of the underlying LLM, we implement **Seeker** using different models, including open-source models and GPT-4. The results are presented in Appendix A.2.4

The results indicate that more advanced language models like GPT-4o lead to better performance in **Seeker**. This suggests that the capabilities of the underlying LLM significantly affect the overall performance of our method.

## 4.6 RQ5: Impact of Domain-Specific Knowledge Integration

To assess the impact of integrating domain-specific knowledge, we compare **Seeker** with and without the inclusion of the Common Exception Enumeration (CEE). The results are shown in Table 3.

Table 3: Impact of Integrating Common Exception Enumeration (CEE)

| Configuration | ACRS | COV (%) | COV-P (%) | ACC (%) | ES | CRS (%) |
|---|---|---|---|---|---|---|
| **Seeker with CEE** | **0.85** | **91** | **81** | **79** | **0.64** | **92** |
| Seeker without CEE | 0.38 | 48 | 41 | 32 | 0.29 | 46 |

The inclusion of CEE leads to significant improvements across all metrics. This demonstrates that integrating domain-specific knowledge enhances **Seeker**'s ability to accurately detect and handle exceptions.

## 4.7 Additional Analysis

In addition to the main experiments, we evaluated **Seeker**'s performance on generating repository-level code and optimizing code patches for GitHub issues. The details of these experiments are provided in Appendix A.3. The results confirm that **Seeker** maintains competitive performance in model-based code generation tasks, further highlighting its robustness and applicability in real-world scenarios.

Our experiments demonstrate that **Seeker** achieves state-of-the-art performance in exception handling code generation. By effectively combining comprehensive exception knowledge with a specialized agent framework, our method addresses the complexities of exception handling in code generation. The superior performance across all metrics highlights the importance of integrating domain-specific knowledge and best practices into code generation models.

## 5 CONCLUSION

In this paper, we extend the study of the impact of prompt specifications on the robustness of LLM generated code. We conduct extensive comparative experiments using four sets of prompt settings and further confirm the mitigating effect of developers' poor exception handling practices. To exploit this phenomenon, we introduce the Seeker method, a multi-agent collaboration framework that provides LLM with the prompt information required for mitigation effects with the support of CEE documents and Deep-RAG algorithms. The upper bound model achieves SOTA performance on exception handling tasks. In general, Seeker can be integrated into any base model, extended to multiple programming languages, and even generalized to knowledge analysis and reasoning of general inheritance relations, such as requirements engineering in Appendix A.3. We hope that our findings and proposed methods can provide new insights and promote future research in these areas. The source code of this paper is available at `https://anonymous.4open.science/r/Seeker`.

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

# A APPENDIX

## A.1 METHOD DETAILS

### A.1.1 DEEP-RAG ALGORITHM

---

**Algorithm 2:** Deep Retrieval-Augmented Generation (Deep-RAG)

---

**Input:** Knowledge hierarchy tree $T$, unit summary $F_i$, detected queries $Q_i$, environment context $Env$

**Output:** Relevant information retrievals $R_i$

1 Initialize relevant knowledge branches set $B = \{\}$;
2 Assign knowledge scenario labels $L = \{l_1, l_2, \dots\}$ to branches of $T$;
3 **foreach** *query $q_{ik}$ in $Q_i$* **do**
4      Identify branches $B_{ik}$ in $T$ related to $q_{ik}$ based on labels $L$;
5      $B \leftarrow B \cup B_{ik}$;

6 **foreach** *branch $b_m$ in $B$* **do**
7      `// Verification Step`
     Select few-sample document examples $X_m = \{x_{m1}, x_{m2}, \dots\}$ associated with branch $b_m$;
8      **foreach** *example $x_{mj}$ in $X_m$* **do**
9          Perform query matching to obtain pass rate $p_{mj}$ and capture accuracy $a_{mj}$;
10          **if** *$p_{mj}$ or $a_{mj}$ below threshold $\theta$* **then**
11              Record failure pattern $fp_{mj}$ based on $Env$;
12              Update environment context $Env$ with $fp_{mj}$;
13      Compute average pass rate $\bar{p}_m$ and accuracy $\bar{a}_m$ for branch $b_m$;
14      **if** *$\bar{p}_m$ or $\bar{a}_m$ below threshold $\theta$* **then**
15          Fine-tune labels $L$ for branch $b_m$ based on aggregated feedback from $Env$;

16 Initialize information retrievals set $R_i = \{\}$;
17 **foreach** *branch $b_m$ in $B$* **do**
18      Select depth level $D$ for node evaluation;
19      **for** $d = 1$ *to $D$* **do**
20          **foreach** *node $n_{ml}$ at depth $d$ in branch $b_m$* **do**
21              Evaluate relevance score $r_{ml}$ to summary $F_i$ and queries $Q_i$;
22              **if** $r_{ml} > \delta$ **then**
23                  Retrieve information $r_{ml}$ from knowledge base;
24                  $R_i \leftarrow R_i \cup \{r_{ml}\}$;

---

In the Deep-RAG algorithm, we assign development scenario labels to each branch of the exception inheritance tree based on their inheritance relationships, enabling the identification of branches that may correspond to specific information of fragile code segments. Acting as an intelligent agent, the algorithm interacts dynamically with its operational environment by leveraging feedback from detection pass rates and capture accuracies obtained during the few-shot verification step. This feedback mechanism allows the system to refine the granularity and descriptions of the scenario labels through regularization prompts derived from failed samples. As a result, Deep-RAG can accurately identify the risk scenarios where fragile codes are located and the corresponding knowledge branches that are activated. Subsequently, the algorithm selectively performs node evaluations on these branches by depth, thereby enhancing retrieval performance and optimizing computational overhead. Additionally, we have designed the algorithm interface to be highly general, ensuring its applicability across a wide range of RAG scenarios beyond exception handling. This generality allows Deep-RAG to support diverse applications, as further detailed in Appendix A.3. By integrating environmental feedback and maintaining a flexible, agent-based interaction model, Deep-RAG not only improves retrieval accuracy and efficiency but also adapts seamlessly to various domains and information retrieval tasks, demonstrating its versatility and robustness in enhancing the performance of large language models.

### A.1.2 COMMON EXCEPTION ENUMERATION

In this section, we introduce the framework for constructing the CEE, which serves as a foundational resource for enhancing the reliability of exception handling in code generation by developers. Without a comprehensive and standardized document like CEE, developers may struggle to accurately detect and handle these exceptions, leading to either overly generic or improperly specific exception management. CEE addresses these challenges by providing a structured and exhaustive repository of exception information, encompassing scenarios, properties, and recommended handling strategies for each exception type. The construction of CEE is guided by three essential rules, each aimed at addressing the complexities of exception management within Java development. First and foremost, we establish a robust standard documentation base, drawing from the Java Development Kit (JDK) to identify and compile a comprehensive set of exception nodes and their descriptions. This foundational layer comprises a total of 433 nodes, organized into 62 branches and spanning five layers within the Java exception hierarchy. By utilizing the standardized documentation from the JDK, we ensure that the CEE is grounded in official, authoritative sources, providing a reliable reference point for exception handling practices. Next, we enhance the CEE by integrating insights from real-world human practices. This involves gathering a range of resources, including enterprise-level Java development documentation and analyzing mature open-source Java projects hosted on platforms like GitHub. By examining exemplary Java code, particularly focusing on effective exception handling practices, we can enrich each exception node in the CEE with detailed contextual information. Specifically, we define three key components for each exception node: **Scenario**, **Property**, and **Handling Logic**.

- **Scenario**: This component describes the specific coding situations or environments in which an exception is likely to occur. By analyzing real-world applications and common coding patterns, we can create realistic scenarios that help developers understand when to anticipate particular exceptions. This contextual understanding is critical for effective exception handling, as it allows developers to write more accurate and responsive code.

- **Property**: This aspect outlines the characteristics and attributes of each exception. Understanding the properties of an exception, such as its severity, possible causes, and the context of its occurrence, they are vital for appropriate handling. This detailed information allows developers to make informed decisions on how to respond to exceptions based on their inherent properties.

- **Handling Logic**: For each exception node, we define best practices for handling the exception. This includes recommended coding strategies, such as specific try-catch blocks, logging mechanisms, and fallback strategies. By incorporating proven handling logic derived from both successful enterprise practices and open-source contributions, we provide a comprehensive guide that assists developers in implementing effective exception management.

The third rule emphasizes the need for fine-grained control over the matching and handling of exceptions through the use of few-shot samples. To ensure that the CEE maintains high accuracy in matching exceptions with the appropriate handling logic, we establish a testing framework comprising a variety of small-scale testing libraries. These libraries are designed to cover a wide range of exceptions, providing high coverage rates for various scenarios. We leverage the CEE in conjunction with these testing libraries to conduct detailed evaluations of exception matching. By analyzing the performance of the CEE in identifying and matching exceptions, we can identify instances of false positives (incorrect matches) and false negatives (missed matches). Based on this analysis, we iteratively refine the information associated with each exception node, adjusting the granularity of the descriptions until we achieve a high accuracy in matching rates. This continuous feedback loop allows us to optimize the CEE for real-world application, ensuring that developers can rely on it to provide accurate and contextually relevant exception handling guidance. By adhering to these rules, the CEE is positioned as a powerful resource that enhances the quality of exception handling in code generated by LLMs. The combination of authoritative documentation from the JDK, insights from real-world practices, and rigorous testing mechanisms creates a comprehensive framework that not only improves the robustness of generated code but also empowers developers with the knowledge and tools they need to manage exceptions effectively. It is worth mentioning that CEE, as a knowledge base, has the value of free expansion and supporting community contributions. We will

continue to be responsible for the version updates and iterations of CEE. An excerpt sample of CEE can be found in Appendix A.2.2

## A.2 EXPERIMENTAL DETAILS

### A.2.1 DATASETS

To ensure the quality and representativeness of the dataset, we carefully selected projects on GitHub that are both active and large in scale. We applied stringent selection criteria, including the number of stars, forks, and exception handling repair suggestions in the project (Nguyen et al., 2020b), to ensure that the dataset comprehensively covers the exception handling practices of modern open-source projects. By automating the collection of project metadata and commit history through the GitHub API, and manually filtering commit records related to exception handling, we have constructed a high-quality, representative dataset for exception handling that provides a solid foundation for evaluating Seeker.

Table 4: The Excerpt Data source

| Repo | Commits | Stars | Forks | Issue Fix | Doc | Under Maintenance |
|------|---------|-------|-------|-----------|-----|-------------------|
| Anki-Android | 18410 | 8500 | 2200 | 262 | Y | Y |
| AntennaPod | 6197 | 6300 | 1400 | 295 | Y | Y |
| connectbot | 1845 | 2480 | 629 | 321 | N/A | Y |
| FairEmail | 30259 | 3073 | 640 | N/A | Y | Y |
| FBReaderJ | 7159 | 1832 | 802 | 248 | Y | N/A |
| FP2-Launcher | 1179 | 25 | 2 | 16 | Y | N/A |
| NewsBlur | 19603 | 6800 | 995 | 158 | Y | Y |
| Launcher3 | 2932 | 91 | 642 | 2 | N/A | Y |
| Lawnchair-V1 | 4400 | 93 | 43 | 394 | Y | Y |
| MozStumbler | 1727 | 619 | 212 | 203 | Y | N/A |

We quantify the quality of datasets in the context of code generation and exception handling using multiple dimensions, encompassing project popularity, community engagement, codebase quality, security posture, documentation integrity and dynamic maintenance. To provide a holistic assessment, we propose a Composite Quality Metric (CQM) that aggregates these dimensions into a single quantitative indicator. Open source code repositories that perform well under this metric enter our semi-automated review process to screen high-quality exception handling blocks for few-shot, CEE building, or testing.

To avoid data leakage, we also performed a round of variations on the test set. Considering that our method does not directly rely on data but fully utilizes the LLM's ability to understand and reason about code, the evaluation results are consistent with our predictions, and the impact of data leakage on the credibility of our method is negligible.

### A.2.2 PROMPT AND DOCUMENT

---

**CEE Prompt Template**

genscenario = """"Below is a kind of exception in java. Please according to the sample discription of scenario of errortype, provide a scenario description of the exception in java just like the sample description.Please note that the granularity of the scenario descriptions you generate should be consistent with the examples.

[Sample Description]
{*sample_desc*}

[Exception]
{*ename*}

---

Note you should output in the json format like below, please note that the granularity of the scenario descriptions you generate should be consistent with the examples:
{{
    "scenario": ...
}}
"""

genproperty = """"Below is a kind of exception in java and its scenario description. Please according to the sample discription of scenario and property of errortype, provide a property description of the exception in java just like the sample description. You can alse adjust the given scenario description to make them consistent. Please note that the granularity of the property descriptions you generate should be consistent with the examples.

[Sample Description]
{sample_desc}

[Exception]
{ename}

[Scenario Description]
{scenario}

Note you should output in the json format like below, please note that the granularity of the property descriptions you generate should be consistent with the examples:
{{
    "scenario": ...;
    "property": ...
}}
"""

---

**Planner Prompt Template**

planner_prompt = """"You are a software engineer tasked with analyzing a codebase. Your task is to segment the given codebase into manageable units for further analysis. The criteria for segmentation are:
- Each unit should have a length within 200 lines.
- The function nesting level should be low.
- The logical flow should be clear and self-contained.
- The segment should be complete and readable.

Given the following codebase:

[Codebase]
{codebase}

Please segment the codebase into units and list them as:

Unit 1:[Code Segment]
{{unit1}}

Unit 2:[Code Segment]
{{unit2}}
...

Ensure that each unit complies with the criteria specified above.
"""

---

**Detector Prompt Template**

detector_senario_match = """"You are a java code auditor. You will be given a doc describe different exception scenarios and a java code snippet.

Your task is to label each line of the code snippet with the exception scenario that it belongs to. If a line does not belong to any scenario, label it with "None". If a line belongs to one of the given scenarios, label it with all the scenarios it belongs to.

[Scenario description]
{*scenario*}

[Java code]
{*code*}

Please output the labeling result in the json format like below:
{{
    "code_with_label": ...
}}
""""

detector_prop_match = """"You are a java code auditor. You will be given a doc describe different exception properties and a java code snippet.

Your task is to label each line of the code snippet with the exception property that it belongs to. If a line does not belong to any property, label it with "None". If a line belongs to one of the given properties, label it with all the properties it belongs to.

[property description]
{*property*}

[Java code]
{*code*}

Please output the labeling result in the json format like below:
{{
  "code_with_label": ...
}}
""""

---

**Predator Prompt Template**

predator_prompt = """"You are a code analysis assistant. Your task is to process the given code unit and identify specific exception types that may be thrown.

[Code Unit]
{*code_unit*}

[Code Summary]
{*code_summary*}

Based on the code summary and the potential exception branches provided, identify the specific exception nodes that may be thrown.

[Potential Exception Branches]
{*exception_branches*}

---

Please answer in the following JSON format:
{{
   "ExceptionNodes": [
      {{
        "ExceptionType": "ExceptionType1",
      }},
      {{
        "ExceptionType": "ExceptionType2",
      }},
      ...
   ]
}}
Ensure that your response strictly follows the specified format.
"""

### Ranker Prompt Template

ranker_prompt = """"You are an exception ranking assistant. Your task is to assign grades to the identified exceptions based on their likelihood and the suitability of their handling strategies.

For each exception, please calculate:

- Exception Likelihood Score (from 0 to 1) based on its attributes and impact.
- Suitability Score (from 0 to 1) of the proposed handling strategy.

[Identified Exceptions and Handling Strategies]
{exception_nodes}

Provide your calculations and the final grades in the following JSON format:
{{
   "Exceptions": [
      {{
        "ExceptionType": "ExceptionType1",
        "LikelihoodScore": value,
        "SuitabilityScore": value,
      }},
      ...
   ]
}}

Please ensure your response adheres to the specified format.

"""

### Handler Prompt Template

handler_prompt = """"You are a software engineer specializing in exception handling. Your task is to optimize the given code unit by applying appropriate exception handling strategies.

[Code Unit]
{code_unit}

[Handling Strategy]
{strategy1}

Generate the optimized code with the applied exception handling strategies.

Please provide the optimized code in the following format:

[Optimized Code]
{{optimized_code}}

Ensure that the code is syntactically correct and adheres to best practices in exception handling.
"""

---

**Sample CEE Node**

```
{
    "name": "IOException",
    "children": [...],
    "info": {
        "definition": "IOException is a checked exception that is thrown when an input-output
operation failed or interrupted. It's a general class of exceptions produced by failed or
interrupted I/O operations.",
        "reasons": "There are several reasons that could cause an IOException to be thrown.
These include: File not found error, when the file required for the operation does not exist;
Accessing a locked file, which another thread or process is currently using; The file system
is read only and write operation is performed; Network connection closed prematurely;
Lack of access rights.",
        "dangerous_operations": "Operations that could typically raise an IOException in-
clude: Reading from or writing to a file; Opening a non-existent file; Attempting to open
a socket to a non-existent server; Trying to read from a connection after it's been closed;
Trying to change the position of a file pointer beyond the size of the file.",
        "sample_code": "String fileName = 'nonexistentfile.txt'; \n FileReader fileReader =
new FileReader(fileName);",
        "handle_code": "String fileName = 'nonexistentfile.txt'; \n try { \n FileReader
fileReader = new FileReader(fileName); \n } catch(IOException ex) { \n Sys-
tem.out.println('An error occurred while processing the file ' + fileName); \n
ex.printStackTrace(); \n }",
        "handle_logic": "Try the codes attempting to establish connection with a
file/stream/network, catch corresponding IOException and report it, output openpath
is suggested."
    },
    "scenario": "attempt to read from or write to a file/stream/network connection",
    "property": "There might be an unexpected issue with accessing the file/stream/network
due to reasons like the file not being found, the stream being closed, or the network
connection being interrupted"
}
```

### A.2.3 COMPUTATION COST ANALYSIS

Integrating a comprehensive exception handling mechanism like **Seeker** introduces potential challenges in computational overhead, especially when dealing with a large number of exception types and complex inheritance relationships. To address this, we designed a high-concurrency interface that keeps the additional computing time overhead constant, regardless of the code volume level. This ensures scalability and controllable complexity when processing any size of codebase.

To evaluate the efficiency of our high-concurrency interface, we conducted experiments on 100 Java code files both before and after implementing parallel processing. For each code file, we executed the exception handling process and recorded the time taken. In the parallelized version, while

the processing between different code files remained sequential, the processing within each code file—specifically, the CEE retrieval involving branch and layered processing—was parallelized.

The results are summarized in Table 5. After applying parallel processing, the average time per code file was reduced to approximately 19.4 seconds, which is about $\frac{1}{15}$ of the time taken with sequential processing. This significant reduction demonstrates the effectiveness of our parallelization strategy.

Table 5: Computation Time Before and After Parallelization

| Processing Method | Average Time per Code File (s) | Speedup Factor |
|---|---|---|
| Sequential Processing | 291.0 | 1x |
| Parallel Processing (Seeker) | 19.4 | 15x |

Notably, the size of the code files did not affect the processing time, indicating that our method efficiently handles codebases of varying sizes without compromising on speed. This stability ensures that **Seeker** can perform consistent and efficient exception handling across any code, making it highly suitable for practical applications.

### A.2.4  FURTHER RESULTS ON DIFFERENT LLMS

We use different open-source (e.g. Code Llama-34B (Rozière et al., 2023), WizardCoder-34B (Luo et al., 2024), Vicuna-13B (Zheng et al., 2023)) and closed-source(e.g. Claude-2 (Clade, 2023), GPT-3-davinci (GPT-3, 2022), GPT-3.5-turbo (GPT-3.5, 2023), GPT-4-turbo (GPT-4, 2023), GPT-4o (GPT-4o, 2024)) LLMs as the agent's internal model to further analyze models' ability for exception handling. The results are summarized in Table 6.

Table 6: Performance of Different Models on Exception Handling Code Generation

| Model | ACRS | COV (%) | COV-P (%) | ACC (%) | ES | CRS (%) |
|---|---|---|---|---|---|---|
| **Open-Source Models** | | | | | | |
| Code Llama-34B | 0.31 | 37 | 35 c | 32 | 0.25 | 34 |
| WizardCoder-34B | 0.37 | 35 | 31 | 29 | 0.28 | 35 |
| Vicuna-13B | 0.23 | 15 | 9 | 11 | 0.19 | 26 |
| **Closed-Source Models** | | | | | | |
| Claude-2 | 0.42 | 64 | 59 | 54 | 0.40 | 54 |
| GPT-3-davinci | 0.56 | 78 | 68 | 60 | 0.48 | 58 |
| GPT-3.5-turbo | 0.63 | 79 | 72 | 66 | 0.52 | 71 |
| GPT-4-turbo | 0.84 | **91** | **83** | 77 | 0.63 | 89 |
| GPT-4o | **0.85** | **91** | 81 | **79** | **0.64** | **92** |

The performance variations among different models can be explained by:

- **Pre-training Data**: Models pre-trained on larger and more diverse code datasets (e.g., GPT-4o) have a better understanding of programming constructs and exception handling patterns.

- **Model Architecture**: Advanced architectures with higher capacities and more layers (e.g., GPT-4) capture complex patterns more effectively.

- **RAG Performance**: Models that efficiently integrate retrieval-augmented generation, effectively utilizing external knowledge (as in our method), perform better.

- **Understanding Capability**: Models with superior comprehension abilities can accurately detect sensitive code regions and predict appropriate exception handling strategies.

Open-source models, while valuable, may lack the extensive training data and architectural sophistication of closed-source models, leading to lower performance. Closed-source models like GPT-4o and GPT-4 benefit from advanced training techniques and larger datasets, enabling them to excel in tasks requiring nuanced understanding and generation of code, such as exception handling.

## A.3   Other Applicable Scenarios Analysis

Figure 6 shows the migration application of Seeker multi-agent framework in APP requirement engineering that also includes parent-child inheritance relationship. We have reason to believe that Seeker framework can try to be compatible with more complex inheritance relationship, being responsible for reasoning representation, while having high performance and interpretability. The above achievements are not easy to accomplish based on graphs or traditional algorithms.

To validate the general applicability of our system in diverse scenarios, we evaluated **Seeker** on standard code generation benchmarks, including **SWE-bench** and **CoderEval**. We present comparative results demonstrating the incremental improvements achieved by our method.

**SWE-bench** is an evaluation framework comprising 2,294 software engineering problems derived from real GitHub issues and corresponding pull requests across 12 popular Python repositories(Jimenez et al., 2024). It challenges language models to edit a given codebase to resolve specified issues, often requiring understanding and coordinating changes across multiple functions, classes, and files simultaneously. This goes beyond traditional code generation tasks, demanding interaction with execution environments, handling extremely long contexts, and performing complex reasoning.

For our experiments, we selected 50 issues related to exception handling from the SWE-bench Lite dataset. Using **GPT-4o** as the internal large model, the **SweAgent**(Yang et al., 2024) coupled with GPT-4o achieved a **19%** *resolve rate* and a **43%** *apply rate*. In contrast, our **Seeker** framework attained a **26%** resolve rate and a **61%** apply rate, indicating a significant improvement.

Table 7: Performance on SWE-bench Lite Exception Handling Issues

| Method | Resolve Rate (%) | Apply Rate (%) |
|---|---|---|
| SweAgent + GPT-4o | 19 | 43 |
| **Seeker** + GPT-4o | **26** | **61** |

**CoderEval** is a benchmark designed to assess the performance of models on pragmatic code generation tasks, moving beyond generating standalone functions to handling code that invokes or accesses custom functions and libraries(Yu et al., 2024). It evaluates a model's ability to generate functional code in real-world settings, similar to open-source or proprietary projects.

In the Java code generation tasks on CoderEval, using **Codex**(Codex, 2021) directly yielded a **Pass@1** score of **27.83%**. When integrating our **Seeker** framework with Codex, the Pass@1 score increased to **38.16%**, demonstrating a substantial enhancement in code generation performance.

Table 8: Performance on CoderEval Java Code Generation Tasks

| Method | Pass@1 (%) |
|---|---|
| Codex | 27.83 |
| **Seeker** + Codex | **38.16** |

These experiments conclusively demonstrate that our **Seeker** framework can achieve significant incremental improvements across different scenarios and benchmarks. By effectively handling exception-related tasks and enhancing code robustness, **Seeker** proves to be a valuable addition to existing code generation models, improving their practical applicability in real-world software engineering problems.

Inspired by OpenAI o1 (o1, 2024) and DoT (Zhang et al., 2024b), we found that Seeker framework has more room for development in LLM reasoning. Through pre-deduction in tree inference, LLM is expected to enter the problem-solving ideas more efficiently and optimize its reasoning actions through interaction with the external environment. In the future, we will continue to explore research in this direction.

## B   RELATED WORK

At present, machine learning has been widely integrated in the field of software engineering, especially in code generation tasks. In this section, we will discuss the progress of Seeker-related work from the latest progress of automatic exception handling tools. These methods have contributed to the robustness or productivity of software engineering, but they also have limitations, which is also the focus of Seeker.

### B.1   AUTOMATIC EXCEPTION HANDLING TOOLS

Zhang et al. (2020) introduced a neural network approach for automated exception handling in Java, which predicts try block locations and generates complete catch blocks in relatively high accuracy. However, the approach is limited to Java and not generalize well without retraining. Additionally, the reliance on GitHub data could introduce biases based on the types of projects and code quality present in the dataset.

Li et al. (2024b) conducted an exploratory study on fine-tuning LLM for secure code generation. Their results showed that after fine-tuning issue fixing commits, the secure code generation rate was slightly improved. The best performance was achieved by fine-tuning using function-level and block-level datasets. However, the limitation of this study is still generalization, not directly applicable to other languages. In addition, it limits the amount and the domain of code that can be effectively processed. Little much code beyond training data scale will affect the processing effect. Li et al. (2023c) also pointed out that in terms of automatic vulnerability detection, the use of traditional fine-tuning methods may not fully utilize the domain knowledge in the pre-trained language model, and may overfit to a specific dataset, resulting in misclassification, excessive false positives and false negatives. Its performance is not as good as emerging methods such as prompt-based learning.

Ren et al. (2023) proposed the Knowledge-driven Prompt Chaining (KPC) approach to improve code generation by chaining fine-grained knowledge-driven prompts. Their evaluation with 3,079 code generation tasks from Java API documentation showed improvements in exception handling. However, the approach's efficiency relies heavily on the inquiry about built-in exceptions for each built-in JDK, and its practical application is limited if the codebase is complex.

Nguyen et al. (2020a) developed FuzzyCatch, a tool for recommending exception handling code for Android Studio based on fuzzy logic. However, the performance of FuzzyCatch depends on the quality and relevance of the training data. In addition, the tool does not perform well for less common exceptions or domains that are not well represented in the training data.

A common limitation of these studies is that the training data they rely on may not fully represent all possible coding scenarios. This may result in a model that is effective in specific situations, but may not generalize well to other situations. In addition, the complexity of exception handling in real-world applications may exceed the capabilities of models trained on more common or simpler cases, so it is crucial to call on the understanding and reasoning capabilities of the model itself. The interpretability of exception handling also provides a guarantee for the improvement of developers' programming literacy. The comparison between the above methods and Seeker is shown in figure 7.

### B.2   MULTI-AGENT COLLABORATION

Multi-agent collaboration refers to the coordination and collaboration between multiple artificial intelligence (AI) systems, or the symbiotic collaboration between AI and humans, working together to achieve a common goal (Smoliar, 1991). This direction has been explored for quite some time (Claus & Boutilier, 1998) (Minsky, 2007). Recent developments show that multi-agent collaboration techniques are being used to go beyond the limitations of LLM, which is a promising trajectory. There are many ways for multi-agents to collaborate with LLM.

VisualGPT (Wu et al., 2023) and HuggingGPT (Shen et al., 2023) explored the collaboration between LLM and other models. Specifically, LLM was used as a decision center to control and call other models to handle more domains, such as vision, speech, and signals. CAMEL (Li et al., 2023a) explored the possibility of interaction between two LLMs. These studies mainly use case studies in the experimental stage to demonstrate their effectiveness and provide specific hints for each case.

For multi-agent collaborative software engineering, which is most relevant to Seeker, Dong et al. (2023) introduces quantitative analysis to evaluate agent collaborative code generation. It introduces the waterfall model in software development methods into the collaboration between LLMs. However, there is still a gap between the evaluation benchmarks used and the actual software development scenarios. In addition, although this work builds a fully autonomous system, adding a small amount of guidance from human experts to supervise the operation of the virtual team will help improve the practicality of the method in actual application scenarios. These problems are exactly what we have improved on Seeker.

Zhang et al. (2024a) formalized the repo-level code generation task and proposed a new agent framework CODEAGENT based on LLM. CODEAGENT developed five programming tools to enable LLM to interact with software artifacts and designed four agent strategies to optimize the use of tools. The experiment achieved improvements on various programming tasks. However, it only integrated simple tools into CODEAGENT. Some advanced programming tools were not explored. This limitation limits the ability of the agent in some challenging scenarios, such as exception handling tasks.

Above all, nowadays, most code-agent works focus on the transformation from the requirements to code and overlook the code robustness during software evolution, which requires not only understanding the requirement but also dealing with potential exceptions.

### B.3 ROBUST SOFTWARE DEVELOPMENT MECHANISM

Code robustness refers to the practices and mechanisms that ensure software to run as expected without causing unexpected side effects, security vulnerabilities, or errors. It involves techniques such as type safety, memory safety, and ensuring that all code paths are well-defined, including when exceptions exist. Exception handling is a necessary programming mechanism to maintain code robustness, allowing programs to manage and respond to runtime errors or other abnormal events. It helps maintain the normal flow of execution and ensures that resources are properly released even when errors occur. Exception handling is critical to code robustness because it ensures that unexpected errors do not compromise the stability or security of the system, prevents resource leaks, ensures data integrity, and keeps the program running correctly even when unforeseen errors occur. (Weimer & Necula, 2004)

From the perspective of code robustness, the defect repair work in the field of software engineering is closely related to exception handling mechanisms, because exception handling involves solving potential errors in the program flow, and developers can mitigate or eliminate defects that may cause program failures or unpredictable behavior.(Jacobs & Piessens, 2009) Currently, since each defect represents a potential vulnerability or instability in the software and is directly related to the functional correctness of the program, research focuses more on defect repair, such as Wen et al. (2023), Devign (Wen et al., 2023), VulAdisor (Wen et al., 2023), while the program's exception safety and exception handling, the powerful program defense mechanisms are not considered.

When a program lacks good exception handling, errors may propagate uncontrollably, leading to resource leakage, data corruption, and potential security vulnerabilities. This situation is called fragile code. After the error occurs, Automatic Program Repair related work performs post-processing to fix the code bug. Representative works include Zhou et al. (2012), Magis (Tao et al., 2024), Huang et al. (2025), PatchFinder (Li et al., 2024d). However, they lack the ability to perceive and repair program risks in advance, and there is a risk of accidentally changing the original function of the code.

**ChatGPT**

**Insensitive Detection**

```
@@ -1,24 +1,28 @@
1   1   /*** Coarse-grained Reminding Prompting
2   2   @param Pay attention to potential exceptions
3   3   */
4   -   public void load(String json) {
    4 + public void load(String json) throws JSONException {
    5 +   mChanged = false;
    6 +   mModels = new HashMap<Long, JSONObject>();
5   7     try {
6   -     mChanged = false;
7   -     mModels = new HashMap<Long, JSONObject>();
8   -     JSONObject modelarray = new JSONObeject(json);
9   -     JSONArray ids = modelarray.names();
    8 +     JSONObject modelArray = new JSONObject(json);
    9 +     JSONArray ids = modelArray.names();
10  10      if (ids != null) {
11  11        for (int i = 0; i < ids.length(); i++) {
12  12          String id = ids.getString(i);
13  -          JSONObject o = modelarray.getJSONObject(id);
    13 +         JSONObject o = modelArray.getJSONObject(id);
14  14          mModels.put(o.getLong("id"), o);
15  15        }
16  16      }
17  17    } catch (JSONException e) {
18  -     System.err.println("JSON processing error: " + e.getMessage());
19  -   } catch (NullPointerException e) {
20  -     System.err.println("Null value encountered: " + e.getMessage());
21  -   } catch (Exception e) {
22  -     System.err.println("Unexpected error: " + e.getMessage());
    18 +     throw new JSONException("Error processing JSON data: " + e.getMessage());
23  19    }
24  20  }
```

**Inaccurate Capture**

```
@@ -1,27 +1,20 @@
1   1   /*** Fine-grained Reminding Prompting
2   2   @param Pay attention to JSONException
3   3   */
4   -   public void load(String json) {
5   -     try {
    4 +   public void load(String json) throws JSONException {
6   5       mChanged = false;
7   6       mModels = new HashMap<Long, JSONObject>();
8   -       JSONObject modelarray = new JSONObject(json);
9   -       JSONArray ids = modelarray.names();
10  -       if (ids != null) {
11  -         for (int i = 0; i < ids.length(); i++) {
12  -           String id = ids.getString(i);
13  -           JSONObject o = modelArray.getJSONObject(id);
14  -           if (o.has("id")) {
    7 +     try {
    8 +       JSONObject modelArray = new JSONObject(json);
    9 +       JSONArray ids = modelArray.names();
    10 +       if (ids != null) {
    11 +         for (int i = 0; i < ids.length(); i++) {
    12 +           String id = ids.getString(i);
    13 +           JSONObject o = modelArray.getJSONObject(id);
15  14          mModels.put(o.getLong("id"), o);
16  -           } else {
17  -             System.err.println("JSONObject for ID " + id + " does not contain 'id' field.");
18  15          }
    16 +       }
    17 +     } catch (JSONException e) {
    18 +       throw new JSONException("Error processing JSON data: " + e.getMessage());
19  19      }
20  -     } catch (JSONException e) {
21  -       System.err.println("JSON processing error: " + e.getMessage());
22  -     } catch (NullPointerException e) {
23  -       System.err.println("Null value encountered: " + e.getMessage());
24  -     } catch (Exception e) {
25  -       System.err.println("Unexpected error: " + e.getMessage());
26  -     }
27  20  }
```

**Incorrect Generating**

```
@@ -1,26 +1,20 @@
1   1   /*** Fine-grained Inspiring Prompting
2   2   @param Pay attention to JSONException. If the JSON data is malformed or not in the expected format, it raises an exception.
3   3   */
4   -   public void load(String json) {
5   -     try {
    4 +   public void load(String json) throws JSONException {
6   5       mChanged = false;
7   6       mModels = new HashMap<Long, JSONObject>();
8   -       JSONObject modelArray = new JSONObject(json);
9   -       if (!modelArray.has("id")) {
10  -         throw new JSONException("The JSON does not contain the expected 'id' field.");
11  -       JSONArray ids = modelArray.names();
12  -       if (ids != null) {
13  -         for (int i = 0; i < ids.length(); i++) {
14  -           String id = ids.getString(i);
15  -           JSONObject o = modelarray.getJSONObject(id);
16  -           mModels.put(o.getLong("id"), o);
    7 +     try {
    8 +       JSONObject modelArray = new JSONObject(json);
    9 +       JSONArray ids = modelArray.names();
    10 +       if (ids != null) {
    11 +         for (int i = 0; i < ids.length(); i++) {
    12 +           String id = ids.getString(i);
    13 +           JSONObject o = modelArray.getJSONObject(id);
    14 +           mModels.put(o.getLong("id"), o);
    15 +         }
17  16        }
    17 +     } catch (JSONException e) {
    18 +       throw new JSONException("Error processing JSON data: " + e.getMessage());
18  19      }
19  -     } catch (JSONException e) {
20  -       System.err.println("JSON processing error: " + e.getMessage());
21  -     } catch (NullPointerException e) {
22  -       System.err.println("Null value encountered: " + e.getMessage());
23  -     } catch (Exception e) {
24  -       System.err.println("Unexpected error: " + e.getMessage());
25  -     }
26  20  }
```

**Good Practice**

```
1   /*** Fine-grained Guiding Prompting
2   @param Pay attention to the JSONException. If you observe which lines of code are prone to the possibility of incorrect JSON data format or not the expected format, try the possible lines together
3   */
4   public void load(String json) throws JSONException {
5     mChanged = false;
6     mModels = new HashMap<Long, JSONObject>();
7     try {
8       JSONObject modelArray = new JSONObject(json);
9       JSONArray ids = modelArray.names();
10      if (ids != null) {
11        for (int i = 0; i < ids.length(); i++) {
12          String id = ids.getString(i);
13          JSONObject o = modelArray.getJSONObject(id);
14          mModels.put(o.getLong("id"), o);
15        }
16      }
17    } catch (JSONException e) {
18      throw new JSONException("Error processing JSON data: " + e.getMessage());
19    }
20  }
```

Figure 5: A schematic diagram of Preliminary Phenomenon, highlight what information will boost LLM & human EH performance, with a case study.

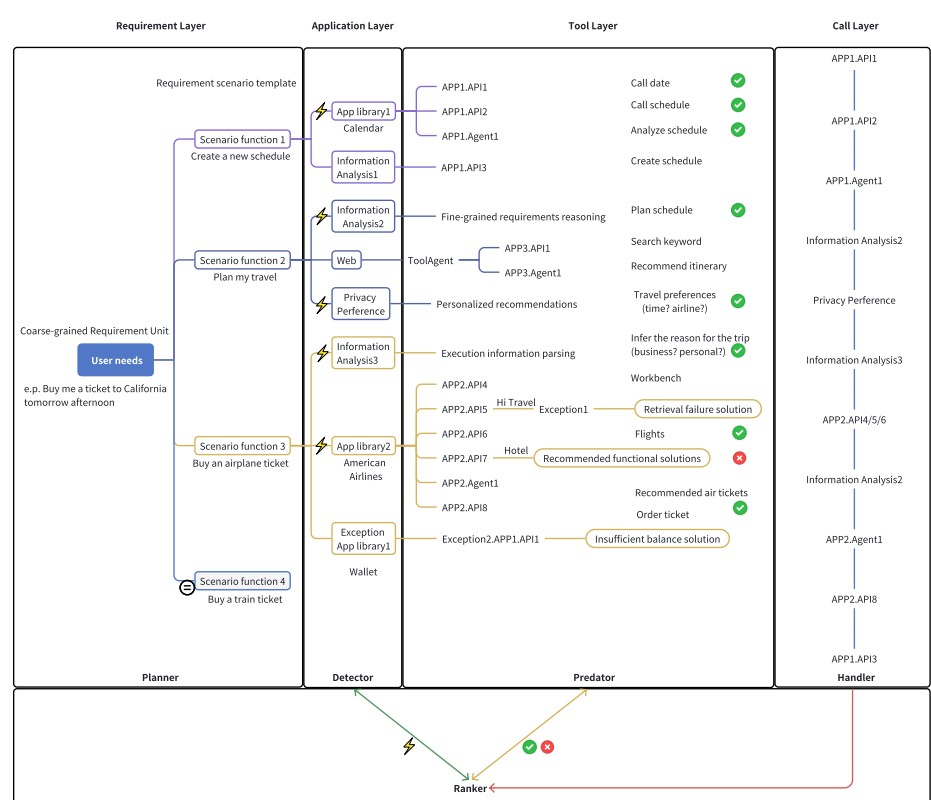

Figure 6: A schematic diagram of APP requirement engineering, highlight seeker's generalizability.

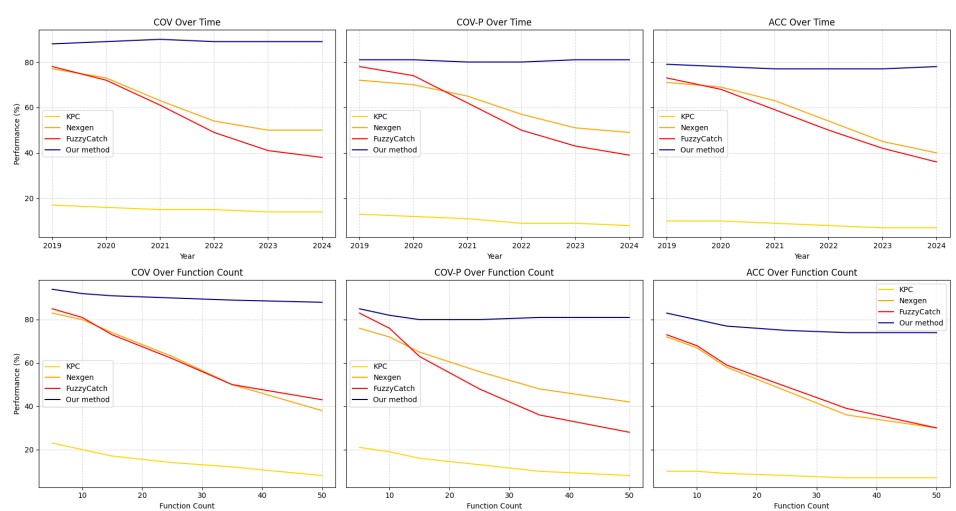

Figure 7: Comparison of Performance Stability Across Baselines and Our Method over Varying Conditions. The top set of curves illustrates the performance metrics over time (2019 to 2024) across different baselines and our method. The bottom set displays performance across increasing function counts.

