# OpenReview forum: "Seeker: Enhancing Exception Handling in Code with a LLM-based Multi-Agent Approach"
_ICLR.cc/2025/Conference — ICLR 2025 Conference Withdrawn Submission_

### Official Review · Reviewer_1fP2 · 2024-11-03

**Soundness:** 2
**Presentation:** 2
**Contribution:** 2
**Rating:** 3
**Confidence:** 2

**Summary:**

The work supports LLMs for exception handling by enhancing them with the seeker framework. The proposed framework has five agents for detecting, capturing and resolving exceptions.  Experimental results indicate that seeker gives better results as compared to other approaches based on multiple measures

**Strengths:**

- The area is interesting and important
- The approach is promising
- An empirical comparison with existing techniques is presented
- The appendix contains details of the method and experiments

**Weaknesses:**

- I found the paper's flow to be quite confusing. It seems the author's had a lot of material to cover, most of which is placed in the appendix. Perhaps because of this, the actual paper lacks clarity and the required detail. It would be helpful if the authors present the material in the main sections, and refer to the appropriate appendix in case the reader wants further detail.
- There seems to be forward referencing in the paper. Material is introduced without proper explanation, and is explained in later sections e.g. Figure1
- The exact contribution(s) need to be written more clearly in the Introduction. Moreover, the material supporting the main contributions seems to be in the appendix and not the main sections e.g. deep-rag algorithm or discussion on the high concurrency.
- The experiments section seems to be defining the evaluation measures rather than focusing on an explanation of the experiments and results
- The authors mention that the superior performance of their approach can be attributed to several factors. However, it is not clear which factor is actually contributing towards the better results
- Some sentences are confusing e.g. in the first para of the Introduction: HumanEval() first proposed to let LLM generating code based on .......

**Questions:**

- The abstract mentions insights for future improvements in code reliability. Where are these discussed?
- The first paragraph of the Introduction discussed the importance of functional correctness of code and it's evaluation. It contains sentences that need to be re-written for describing the existing work more clearly
- What do you mean by the main peak area?
- Where is the research question " Do we need to enhance the standardization, interpretability and generalizability of exception handling in real code development scenarios?" answered?
- The explanation of Figure 1 in the introduction is not clear
- How was the chain of thought in Fig 1b) developed?
- Why is there a need of section 3.1 in Methodology?
- What is Figure 2 showing and where is it referred?
- In algorithm1, where is 'low nesting level' defined. What is meant by 'logical flow is clear'?
- What is meant by the 'thoughtful approach' of the planner?
- Is there a need to dedicate more than a page to defining the evaluation measures in the Experiments section?
- Where are the industry practices that were employed in the approach discussed?

---

> ### Author Response · Authors · 2024-11-16
> **Strong rebuttal to Reviewer 1fP2 (Part 1)**
>
> **Thank you very much for your careful and enthusiastic comments. We will respond to all the questions you raised in your comments and provide easy-to-understand explanations of the issues in combination with the revised version. Considering that some misunderstandings may come from the discussion in the first draft of the paper, or from your lack of understanding of the paper and related issues, we hope that you will fully discuss the rebuttal and the revised version information, give us your advice, and reconsider your score, based on our full confidence in this paper.**
>
> **Strengths:** The area is interesting and important
>
> The approach is promising
>
> An empirical comparison with existing techniques is presented
>
> The appendix contains details of the method and experiments
>
> **Response:** Thank you for your recognition of the motivation, methods, experimental adequacy, and content of this article.
>
> **Weakness1:** I found the paper's flow to be quite confusing. It seems the author's had a lot of material to cover, most of which is placed in the appendix. Perhaps because of this, the actual paper lacks clarity and the required detail. It would be helpful if the authors present the material in the main sections, and refer to the appropriate appendix in case the reader wants further detail.
>
> **Response:** Main Text Structure and Content Decisions:
>
> We understand the importance of making the main text self-contained. In the initial submission, due to page limitations, we prioritized content on motivation, the broader framework, and the overall experimental impact, deferring some algorithmic details to the appendix. However, based on your feedback, we will revise the main text to include essential explanations and examples of CEE and Deep-RAG to ensure a clearer, more integrated understanding.
>
> Contextual Justification for the Appendix: Given the complexity of our method, specific algorithmic details require additional space to avoid oversimplification. The appendix provides extended technical elaboration and decision-making insights, which we view as valuable for those interested in replicating or extending our approach. Our revised structure will provide readers with a self-contained overview while preserving the appendix's detailed role.
>
> **Weakness2:** There seems to be forward referencing in the paper. Material is introduced without proper explanation, and is explained in later sections e.g. Figure1
>
> **Response:** Thank you for your comments. We have improved the readability of the paper to a certain extent in the revised version. We will pay attention to the issues you mentioned and make further improvements in subsequent revisions.
>
> **Weakness3:** The exact contribution(s) need to be written more clearly in the Introduction. Moreover, the material supporting the main contributions seems to be in the appendix and not the main sections e.g. deep-rag algorithm or discussion on the high concurrency.
>
> **Response:** Thank you for the comment. We try to more clearly indicate contributions in the rebuttal revision, including:
>
> 1. We highlight the importance of standardization, interpretability, and generalizability in exception handling mechanisms, identifying a gap in existing research.
>
> 2. We propose Seeker, which decomposes exception handling into specialized tasks and incorporates Common Exception Enumeration (CEE) to enhance performance.
>
> 3. We introduce a deep retrieval-augmented generation (Deep-RAG) algorithm tailored for complex inheritance relationships, improving retrieval efficiency.
>
> 4. We conduct extensive experiments demonstrating that Seeker improves code robustness and exception handling performance in LLM-generated code.
>
> **Weakness4:** The experiments section seems to be defining the evaluation measures rather than focusing on an explanation of the experiments and results
>
> **Response:** Thank you for your insightful comments on the experiment section. In the first draft, we wanted to clarify the motivation for the work. Intuitively, we pointed out that existing common metrics (such as Pass@k, Recall@k) cannot evaluate the quality of exception handling in generated code. Therefore, we tried to clearly introduce the evaluation metrics suitable for exception handling subtasks (fragile code location, exception type capture, exception handling generation). In the revised version, we established and described the mapping of each metric on the Detection-Capture-Handling task relationship in detail, and expanded the evaluation ideas to other tasks of non-functional correctness of the code to illustrate the rationality. We also found the concerns you raised. Considering the content limitations, we have greatly optimized the experimental content in the revised version, added ablation experiments, general metric experiments, experimental settings and result discussions, and optimized the metric representation to intuitively understand its rationality.

---

> ### Author Response · Authors · 2024-11-16
> **Strong rebuttal to Reviewer 1fP2 (Part 2)**
>
> **Weakness5:** The experiments section seems to be defining the evaluation measures rather than focusing on an explanation of the experiments and results
>
> **Response:** 1. Detailed Factor Analysis via Ablation Studies
>
> To clarify the contributions of specific components in our framework, we have conducted ablation experiments, focusing on the effects of each core agent and the Common Exception Enumeration (CEE) knowledge base. As demonstrated in Table 2, the removal of any agent results in measurable performance degradation across metrics such as Automated Code Review Score (ACRS), Coverage Pass (COV-P), and Code Review Score (CRS). These experiments illustrate that each agent—particularly the Detector, Predator, and Handler—plays a significant role in improving both the precision of exception handling and the robustness of code generation.
>
> 2. Evaluation of CEE’s Impact
>
> The inclusion of CEE substantially enhances Seeker’s effectiveness, as shown in Table 3. By providing a comprehensive repository of real-world exception scenarios, properties, and handling logic, CEE enables precise matching of exception types to appropriate handling strategies. This integration is evident in our results, where Seeker with CEE outperforms the version without it across all evaluation metrics, including a significant improvement in Accuracy (ACC) and Edit Similarity (ES). This clearly demonstrates CEE’s critical role in improving exception detection and handling precision.
>
> 3. Comparative Analysis with Baseline Stability
>
> We also compared Seeker’s performance stability against baselines over time and code complexity levels (Appendix A.2.1). Unlike other methods, which exhibit performance variability with changes in code snippet creation time and function complexity, Seeker maintains stable, high-quality exception handling across conditions. This stability highlights the advantage of our structured, agent-based approach in dynamically adapting to code variations.
>
> We hope this explanation clarifies the distinct contributions of each factor to Seeker’s performance.
>
> **Weakness6:** Some sentences are confusing e.g. in the first para of the Introduction: HumanEval() first proposed to let LLM generating code based on .......
>
> **Response:** Thanks for pointing out. We also found this paragraph a little confusing when we reviewed it. However, considering the data settings and format of the HumanEval benchmark itself:
>
> Each piece of data in the HumanEval dataset contains a programming task, with the following structure:
>
> Prompt: Contains a function definition and a docstring describing the task. The docstring describes the function's functionality and the expected format of input and output.
>
> Test Cases: Contains several test cases to verify whether the code generated by the model meets the requirements. Test cases are generally in the form of assert statements in Python.
>
> Solution: Standard reference solution code. This code is a completely correct implementation and can be used as a benchmark for generating code.
>
> We would like to try to summarize the key points here.
>
> **Question1:** The abstract mentions insights for future improvements in code reliability. Where are these discussed?
>
> **Response:** We tried to illustrate the connection between exception handling and code reliability in the first draft of the paper:
>
> [Introuction]
>
> Ren et al. (2023) conducted an in-depth study on the performance of LLM-generated code in code robustness represented by exception handling mechanisms, which opened up new explorations for LLM to predict and handle potential risks of generated code itself before a vulnerability occurs.
>
> ...
>
> Exception handlers that are too general can make code more error-prone by catching and handling exceptions that were not anticipated by the programmer and for which the handler was not intended. Osman et al. (2017) further demonstrates that capturing accurate fine-grained exceptions can help developers quickly identify the source of the problem, effectively improve the readability and maintainability of the code, and avoid mishandling different types of errors. However, due to the lack of good handling paradigm experience for long-tail, domain-specific, or customized exception types, combining with the complex inheritance relationship and the multi-pattern of exception handling, it is still challenging to accurately achieve this goal.
>
> And [A Revisit of Human Empiricals].
>
> From the actual Java software development scenario we are currently targeting, Java relies heavily on exceptions as a mechanism for handling exceptional events. This greatly illustrates the direct connection between exception handling and code robustness. In addition, More code robustness mechanisms have similar characteristics to exception handling, such as input validation and verification, assertions, etc. We believe that Seeker's methodology is scalable in these issues, and exception handling undoubtedly has the most urgent requirement.

---

> ### Author Response · Authors · 2024-11-16
> **Strong rebuttal to Reviewer 1fP2 (Part 3)**
>
> (Continue Comment) In addition, in the revised version, we further discussed code robustness related work, such as defect repair and APR, in the appendix Robust Software Development Mechanism. Further discussion is welcome!
>
> **Question2:** The first paragraph of the Introduction discussed the importance of functional correctness of code and it's evaluation. It contains sentences that need to be re-written for describing the existing work more clearly
>
> **Response:** Thank you for your guidance. We have indeed rewritten the Introduction in the revised version, and you may provide more specific guidance on this point.
>
> **Question3:** What do you mean by the main peak area?
>
> **Response:** 1. Clarification of "Main Peak Area" Concept
>
> The term "main peak area" refers to the distribution pattern of exception types within the inheritance tree, as shown in Figure 2 of our paper. In this context, the "main peak area" signifies clusters of commonly encountered exceptions that lie within the most active or frequently referenced nodes of the exception hierarchy. These nodes represent exception types with a higher occurrence probability in real-world scenarios, as they often capture standard and predictable error events, thus forming a "peak" in the distribution.
>
> 2. Distribution Patterns and Exception Handling Strategy
>
> Figure 2 demonstrates how exception types are aligned within the Java inheritance hierarchy, highlighting the main peak area where more common exceptions are concentrated. This distribution pattern helps us to better manage exceptions by targeting the handling logic more effectively toward higher-risk or frequently encountered exceptions, optimizing detection and response efforts in the "main peak area" and reducing the complexity associated with handling rare, low-probability exception types.
>
> 3. Hierarchical Significance of the Main Peak
>
> As illustrated, the "main peak area" emphasizes the standard exception handling practices required for a substantial portion of Java exceptions that lie within core branches of the inheritance tree. This targeted approach enables our methodology to more accurately capture and handle exceptions at appropriate levels of specificity, leveraging hierarchical relationships effectively to improve exception handling outcomes.
>
> **Question4:** Where is the research question " Do we need to enhance the standardization, interpretability and generalizability of exception handling in real code development scenarios?" answered?
>
> **Response:** *Introduction of the Research Question and Its Answering in Experimental Sections*
>
> The posed research question is addressed through our methodology and results, specifically in the context of improving exception handling standards, interpretability, and generalizability in real-world code development. Section 3.1 introduces the critical importance of interpretable and standardized exception handling, aligning it with the Seeker framework's structured approach to achieve these enhancements.
>
> *Link to Experiment Setup and Findings (RQ4)*
>
> Our experimental setup (Section 4.1) and the evaluation of Seeker’s performance (particularly in Research Question 4, Table 3) explicitly investigate the impact of integrating domain-specific knowledge like the Common Exception Enumeration (CEE) to ensure that exception handling mechanisms are standardized and generalizable across various development scenarios. This setup is pivotal in showcasing that standardized and interpretable exception handling can indeed be systematically achieved.
>
> *Illustrative Evidence Through Case Studies and Visualizations*
>
> We highlight Figure 1(a), which demonstrates the performance impact of various prompting levels on exception handling quality, further addressing the need for enhanced exception handling interpretability. Additionally, the case study results, as seen in Section 4.3, illustrate the positive impact of generalizable exception handling strategies by showcasing how the Seeker framework adapts to complex exception scenarios with high accuracy.
>
> *Implications for Future Development*
>
> The combination of results in these sections answers the research question by providing data-driven evidence that standardizing and generalizing exception handling practices, through structured agent-based methods, significantly enhances code robustness and reliability in real-world scenarios.
>
> **Question5:** The explanation of Figure 1 in the introduction is not clear
>
> How was the chain of thought in Fig 1b) developed?
>
> **Response:** Thank you for your question regarding the development of the "chain-of-thought" illustrated in Figure 1(b). This framework synthesizes concepts from both industry insights and prior research to depict the thought process of senior developers when handling exceptions. The structure of the chain-of-thought model was informed by:

---

> ### Author Response · Authors · 2024-11-16
> **Strong rebuttal to Reviewer 1fP2 (Part 4)**
>
> (Continue Comment) 1. Prior Research in Exception Handling:
>
> Exception Range and Sensitive Code: Prior work, such as Nakshatri et al. (2016) and de Pádua & Shang (2017), emphasizes the importance of identifying specific exception types and capturing lower-level exceptions within the class hierarchy for improved debugging and code robustness. These studies indicate that developers benefit from targeting specific exception types rather than overly broad handlers, which supports the inclusion of “Exception Range” and “Sensitive Code” as core components in the model.
>
> 2. Insights from Industry Consultation:
>
>  To complement the research literature, we consulted with an experienced engineer from Huawei, who provided insights into the practical aspects of exception handling. The expert emphasized the importance of drawing on programming experience and specific knowledge resources (like SDK documentation) to manage complex exception handling cases. This feedback informed the inclusion of “Programming Experience” and “Knowledge Invocation” as part of the developer thought process.
>
> 3. Key Terms in Our Experimental Setup:
>
>  Each term in the figure directly links to elements of our experiments and module design. Exception Matching and Handling Design with cues from our Common Exception Enumeration (CEE) document is our basic logic of generalized information. “Grammar Mastery” and “SDK Learning” help reinforce accurate exception targeting, while “Handling Method” and “Resolution Effects” provide structured guidance on selecting and evaluating exception handling strategies.
>
> 4. Clarification on the Model's Purpose:
> While these elements may appear as standalone terms, they represent structured stages in a senior developer’s approach to handling exceptions, as supported by empirical studies and industry practices. We can enhance Figure 1(b) to make these relationships clearer, showing how each component fits into a systematic workflow rather than a list of separate “buzzwords.”
>
> **Question6:** Why is there a need of section 3.1 in Methodology?
>
> **Response:** Rationale for Pre-experiment Justification
>
> Section 3.1 serves as a foundation for understanding the necessity and validity of our pre-experiments, specifically focusing on establishing robust exception handling mechanisms within our framework. By delineating "Rules of Good Practice," we can clarify the basis on which fragile code segments are managed and enhance the accuracy of exception handling, which directly impacts the Seeker model's effectiveness. This segment validates our methodological approach by contextualizing exception handling as a structured process, reinforcing the credibility of our experimental design and results.
>
> Clarification of Figure 1(b)
>
> As elaborated in Section 3.1, Figure 1(b) represents the logical process followed by expert developers when handling exceptions, a process our model aims to replicate. This schematic diagram is essential for explaining how the preliminary prompt settings (e.g., Coarse-grained and Fine-grained prompting) impact exception handling. The section thus provides readers with an interpretive lens for evaluating Figure 1(b), allowing a better understanding of the improvement Seeker brings by mirroring this step-by-step developer expertise.
>
> **Question7:** What is Figure 2 showing and where is it referred?
>
> **Response:** 1. Clarification of "Main Peak Area" Concept
> The "main peak area" signifies clusters of commonly encountered exceptions that lie within the most active or frequently referenced nodes of the exception hierarchy. These nodes represent exception types with a higher occurrence probability in real-world scenarios, as they often capture standard and predictable error events, thus forming a "peak" in the distribution.
>
> 2. Distribution Patterns and Exception Handling Strategy
>
> Figure 2 demonstrates how exception types are aligned within the Java inheritance hierarchy, highlighting the main peak area where more common exceptions are concentrated. This distribution pattern helps us to better manage exceptions by targeting the handling logic more effectively toward higher-risk or frequently encountered exceptions, optimizing detection and response efforts in the "main peak area" and reducing the complexity associated with handling rare, low-probability exception types.
>
> 3. Hierarchical Significance of the Main Peak
>
> As illustrated, the "main peak area" emphasizes the standard exception handling practices required for a substantial portion of Java exceptions that lie within core branches of the inheritance tree. This targeted approach enables our methodology to more accurately capture and handle exceptions at appropriate levels of specificity, leveraging hierarchical relationships effectively to improve exception handling outcomes.

---

> ### Author Response · Authors · 2024-11-16
> **Strong rebuttal to Reviewer 1fP2 (Part 5)**
>
> **Question8:** In algorithm1, where is 'low nesting level' defined. What is meant by 'logical flow is clear'?
>
> **Response:** *Definition of “Low Nesting Level”*
>
> In our revised manuscript, we clarify that "low nesting level" refers to the structural complexity of the code. Specifically, it denotes segments with a limited depth （we set it to 5） of nested functions or conditional statements, thus ensuring simplicity and minimizing intricate dependencies. As defined in Algorithm 1 (Lines 3–4), this criterion is applied during the segmentation process to maintain manageable, straightforward code units.
>
> *Explanation of “Logical Flow is Clear”*
>
> We use "logical flow is clear" to describe segments that possess a self-contained and coherent flow of logic, avoiding excessive branching or non-linear control structures. This criterion helps in preserving the readability and interpretability of code units, as noted in the segmentation criteria in Algorithm 1.
>
> *Alignment with Code Structuring Goals*
>
> Our segmentation criteria, including low nesting level and clear logical flow, aim to facilitate efficient analysis and handling by the Seeker framework. We have added further explanations in the revised manuscript to clarify these terms in the context of code segmentation and exception handling (see Section 3.2 and Appendix A.1).
>
> **Question9:** What is meant by the 'thoughtful approach' of the planner?
>
> **Response:** *Definition of the Planner's Approach*
>
> The "thoughtful approach" refers to the planner agent's strategy for segmenting code into manageable units, balancing critical factors such as code volume, dependency levels, and requirement relationships. As clarified in Section 3.2 of our paper, this segmentation helps optimize the process by minimizing complexity without disrupting the logical structure of the code.
>
> *Purpose and Benefits of this Approach*
>
> The planner's "thoughtful" method prioritizes context window constraints and avoids excessive fragmentation, ensuring each unit remains coherent and contextually complete. This approach prevents overwhelming downstream agents, supporting effective and efficient processing within the Seeker framework. This segmentation approach is detailed in the Methodology section, highlighting its role in maintaining clarity while addressing complex dependencies.
>
> *Supporting Evidence*
>
> As described in Section 3.2, this segmentation strategy enables Seeker to handle larger codebases effectively, maintaining performance across diverse code scenarios. By avoiding overly fine divisions, we ensure both stability and scalability when addressing complex codebases, ultimately enhancing code robustness.
>
> **Question10:** Is there a need to dedicate more than a page to defining the evaluation measures in the Experiments section?
>
> **Response:** Same response to **Weakness4**.
>
> **Question11:** Where are the industry practices that were employed in the approach discussed?
>
> **Response:** *Absence of Industry-Wide Standards for Exception Handling*
>
> Currently, there is no widely adopted, standardized approach within the software industry for addressing exception handling comprehensively. As described in our paper, exception handling practices are often left to individual development teams, relying on ad hoc guidelines tied to immediate operational stability rather than unified standards. This gap is among the core motivations for our work with Seeker, as we aim to offer a structured, generalized approach to exception handling.
>
> *Limitations of Conventional Exception Practices*
>
> Traditional practices in exception handling generally center around ensuring basic system stability or addressing specific bugs, often neglecting the broader potential of exception handling to enhance code robustness. As noted in Section 2.2, these conventional methods usually satisfy minimal functional requirements but fail to leverage the full capacity of exceptions to safeguard application resilience in a structured manner.
>
> *Contribution of Our Work*
>
> Our work addresses this industry gap by introducing a framework that systematically enhances exception handling practices. As detailed in Section 3.2, we propose the Common Exception Enumeration (CEE) and the Seeker multi-agent framework, which offer best practices derived from multiple sources, including extensive high-quality exception documentation and developer guidelines. This framework supports developers in achieving robust exception handling by providing comprehensive guidance based on industry-aligned, real-world scenarios rather than isolated project-specific practices.
>
> **Above all, thank you very much for your comment!  We hope that you (here refers to PC, reviewers) can comprehensively consider the motivation, method details, and experimental results, and reconsider/give objective review results.**

---

### Official Review · Reviewer_QC3A · 2024-11-04

**Soundness:** 3
**Presentation:** 4
**Contribution:** 4
**Rating:** 5
**Confidence:** 4

**Summary:**

This paper addresses a critical issue in software development: the deficiencies in exception handling practices that lead to unreliable and fragile code. The authors propose a novel multi-agent framework called Seeker, which leverages large language models (LLMs) to enhance exception handling in programming, specifically within Java codebases. Seeker mainly has five dedicated agents: *Planner* to segment a given codebase into manageable units, *Detector* to identify fragile areas in the code, *Predator* to incorporate the external knowledge, *Ranker* to assign grades to the detected exceptions, and *Handler* to generate optimized code incorporating exception-handling strategies. Accordingly, this paper systematically identifies common pitfalls in exception handling and proposes solutions based on empirical studies of human developer practices. The empirical evaluation shows that combining comprehensive exception knowledge with a specialized agent framework as in Seeker, exception-handling code can be generated with adequate success.

**Strengths:**

- Presents the first systematic study of LLMs in the context of exception handling, which can better scale than all related work, to real-world scenarios.
- The integration of a Common Exception Enumeration (CEE) suggests that the framework can be widely adopted by developers, enhancing the robustness of code generation practices.
- The paper is well-written and easy to follow.

**Weaknesses:**

- *Missing related work / potential baseline* [1]: Exception-handling code is highly project-specific. Accordingly, the cited work deals with the tasks of: (1) identifying code that can throw exceptions, (2) localizing the statements that belong to the try-block, and (3) identifying the exceptions the highlighted block can throw (everything but the objective of *Handler* in *Seeker*).
- *Limited Discussion on Evaluation Granularity*: Evaluation primarily focuses on overall performance metrics without a deeper discussion on the granularity of exception handling improvements. For instance, some code can throw multiple exceptions, or have multiple blocks that need handling. Further insights into specific types of exceptions and their handling would strengthen the findings.
- *Limited evaluation*: Gap between experiments and final conclusions drawn.


[1] Yuchen Cai, Aashish Yadavally, Abhishek Mishra, Genesis Montejo, and Tien Nguyen. 2024. Programming Assistant for Exception Handling with CodeBERT. In Proceedings of the IEEE/ACM 46th International Conference on Software Engineering (ICSE '24). Association for Computing Machinery, New York, NY, USA, Article 94, 1–13. https://doi.org/10.1145/3597503.3639188

**Questions:**

1. How might the framework be adapted to evaluate exception handling at a more granular level?
2. How do the authors ensure that the handling strategies recommended by the Handler agent are not only correct but also optimal?

---

> ### Author Response · Authors · 2024-11-16
> **Rebuttal to Reviewer QC3A (Part 1)**
>
> **Thank you very much for your careful comments. We will respond to all the questions you raised in your comments and provide easy-to-understand explanations of the issues in combination with the revised version. Considering that some misunderstandings may come from the discussion in the first draft of the paper and your temporary confusion at the time, we hope that you will fully discuss the rebuttal and the revised version information, give us your advice, and reconsider your score, based on our full confidence in this paper.**
>
> **Strengths1:** Presents the first systematic study of LLMs in the context of exception handling, which can better scale than all related work, to real-world scenarios.
>
> **Response:** Thank you for your recognition of this article's contribution and experimental results.
>
> **Strengths2:** The integration of a Common Exception Enumeration (CEE) suggests that the framework can be widely adopted by developers, enhancing the robustness of code generation practices.
>
> **Response:** Thank you for recognizing the contribution, promise, and potential of this article.
>
> **Strengths3:** The paper is well-written and easy to follow.
>
> **Response:** Thank you for your recognition of the writing quality of this article.
>
> **Weakness1:** Missing related work / potential baseline [1]: Exception-handling code is highly project-specific. Accordingly, the cited work deals with the tasks of: (1) identifying code that can throw exceptions, (2) localizing the statements that belong to the try-block, and (3) identifying the exceptions the highlighted block can throw (everything but the objective of Handler in Seeker).
>
> **Response:** Thank you for your subtasks baseline suggestion. Well, combined with our research and the related work you provided, I don't think there is enough research work to split the exception handling task into three fine-grained subtasks and compare them with the baseline. Combined with related work B1 and the traditional methods of exception handling, first of all, although pure static analysis can be implemented separately, its effect and cost have been criticized. From the perspective of improving the effect, the interpretability of static analysis conflicts with the uninterpretability of neural networks. Therefore, there has been little work related to static analysis from the perspective of effect in recent years. As for the pre-training or fine-tuning method, we know that this is a direct i/o. All we can do is to divide the indicators from their output results to evaluate the subtasks. As we did, The indicators of ACRS and CRS correspond to the overall exception handling quality, COV corresponds to the performance of Detector (identifying code that can throw exceptions), COV-P corresponds to the performance of Predator selection (localizing the statements that belong to the try-block), ACC corresponds to the performance of Ranker's exception type selection (identifying the exceptions the highlighted block can throw), and ES corresponds to the correlation of exception handling strategies compared with golden.
>
> **Weakness2:** Limited Discussion on Evaluation Granularity: Evaluation primarily focuses on overall performance metrics without a deeper discussion on the granularity of exception handling improvements. For instance, some code can throw multiple exceptions, or have multiple blocks that need handling. Further insights into specific types of exceptions and their handling would strengthen the findings.
>
> **Response:** Thank you very much for your comments on the evaluation granularity. You are very professional, but I think we can cover and discuss this area. One of our core contributions is that the Deep-RAG algorithm based on path activation is capable of covering multi-pattern exception handling.
>
> 1. When the try block should throw multiple exceptions: This means that multiple exception paths are activated.
>
> 2. When the try block should have nested exception handling: This means that the Detector-Predator combination is triggered continuously.
>
> 3. When exception handling should be performed in the catch block: If the exception type has this common requirement, the exception handling strategy will include this suggestion and be executed by the Handler.
>
> In addition, our evaluation method can cover the evaluation of multi-pattern, see the response to Weakness1 and Section 4.1.3 of the paper for details.
>
> **Weakness3:** Limited evaluation: Gap between experiments and final conclusions drawn.
>
> **Response:** Thank you for your evaluation of the completeness of our first draft experiment. In the first draft version, we hope to explain the motivation of the work more clearly. Intuitively, we pointed out that the existing common indicators (such as Pass@k, Recall@k on dependencies) cannot evaluate the quality of exception handling of generated code.

---

> ### Author Response · Authors · 2024-11-16
> **Rebuttal to Reviewer QC3A (Part 2)**
>
> (Continue Comment) Therefore, we tried to clearly introduce the evaluation indicators suitable for exception handling subtasks (fragile code location, exception type selection, exception handling generation) in the first draft. In the revised version, we established and described the mapping of each indicator in the Detection-Capture-Handling task relationship in detail, and expanded the evaluation ideas to other tasks of non-functional correctness of the code to illustrate the rationality. We also found the concerns you raised. Considering the content limitations, we have greatly optimized the experimental content in the revised version, added ablation experiments, general indicator experiments, experimental settings and results discussions, and optimized the indicator representation to intuitively understand its rationality.
>
> **Weakness4:** [1] Yuchen Cai, Aashish Yadavally, Abhishek Mishra, Genesis Montejo, and Tien Nguyen. 2024. Programming Assistant for Exception Handling with CodeBERT. In Proceedings of the IEEE/ACM 46th International Conference on Software Engineering (ICSE '24). Association for Computing Machinery, New York, NY, USA, Article 94, 1–13. https://doi.org/10.1145/3597503.3639188
>
> Response: Thank you for providing the related work. We carefully read the paper and put forward the following analysis on the similarities, differences, advantages and disadvantages of our work:
>
> Neurex is the first neural-network model to automated exception handling recommendation in three tasks for (in)complete code. It is designed to capture the basic insights to overcome key limitations of the state-of-the-art IR approaches. With the learning-based approach, it does not rely on a pre-defined threshold for explicit feature matching.
>
> Neurex still has several limitations. First, it cannot generate new exception types that were not in the training corpus with low cost, but Seeker can cover this issue with revising CEE.
>
> Second, it does not support the generation of exception handling code inside the catch body. Each project might have a different way to handle exception types in the catch body. Training methods are hard to cover various exceptional handling strategies. While using Seeker, those handling strategies are contained in our CEE suggestions, and reasonable applications and developer expansion are obtained through Ranker.
>
> Third, Neurex needs training data, thus, does not work for a new library without any API usage yet.
>
> Fourth, it is possible that the model produces the labels O, B-Try, and I-Try that do not correspond to the legal way of a try-catch block.
>
> Fifth, XBlock might produce a conflicting result with XState.
>
> Sixth, Neurex is specifically for Java, and was only validated on a balanced dataset based on Android and JDK libraries. The results might vary for other libraries, and might not be representative, and might not reflect well the ratio in practice.
>
> While Seeker avoids those limitations basing on Deep-RAG, effectively.
>
> **Question1:** How might the framework be adapted to evaluate exception handling at a more granular level?
>
> Response: *Clarification on Multi-Pattern Evaluation Coverage (W2)*
>
> Our framework’s multi-pattern evaluation already enables a nuanced analysis of exception handling by accounting for diverse handling patterns across different exception types and scenarios. As outlined in Section 3.2, we emphasize a detailed examination of exception handling practices, leveraging distinct metrics that evaluate robustness and adherence to coding standards. Specifically, the use of Coverage (COV) and Coverage Pass (COV-P) metrics in our framework provides an accurate measure of our method's ability to identify and appropriately manage various sensitive code segments, capturing the depth and specificity of the exception handling practices implemented.
>
> *Granularity Through Agent-Based Detection and Ranking*
>
> Our framework’s agents, particularly the Detector and Ranker agents, are designed to ensure granular control and evaluation of exception handling by detecting subtle variations in exception types and assigning scores based on likelihood and suitability (see Section 4.3 for ablation studies). This multi-agent approach allows us to dissect exception handling practices into specific, context-sensitive responses. As discussed in Section 3.1, this level of granularity ensures that the framework captures fine-grained exception handling patterns and assesses them in alignment with best practices.

---

> ### Author Response · Authors · 2024-11-16
> **Rebuttal to Reviewer QC3A (Part 3)**
>
> (Continue Comment) *Enhanced Metric Detail in Future Revisions*
>
> We acknowledge that further detailing these metrics, especially Coverage and Accuracy, to reflect finer distinctions in exception handling strategies could add to the granularity of evaluation. Expanding these metrics to distinguish between frequently occurring versus long-tail exceptions would further enhance our ability to assess complex, multi-pattern scenarios. We are committed to refining our metrics in future work to address the layers of complexity in exception handling practices across diverse development scenarios.
>
> **Question2:** How do the authors ensure that the handling strategies recommended by the Handler agent are not only correct but also optimal?
>
> **Response:** *Ensuring Correctness through CEE Integration*
>
> The recommended handling strategies are grounded in the Common Exception Enumeration (CEE), which we developed by compiling comprehensive documentation from trusted sources such as the Java Development Kit (JDK) and best practices from industry standards. The CEE provides well-defined scenarios, properties, and handling logic for each exception type, ensuring that the Handler’s strategies are accurate and aligned with established practices. This structured and exhaustive repository equips the Handler agent with authoritative and contextually appropriate handling guidelines, which are tailored to each specific exception type and scenario.
>
> *Optimizing Strategies via the Ranker Agent*
>
> To further ensure optimality, we employ a Ranker agent that evaluates each handling strategy on a graded scale, factoring in both the likelihood of exception occurrence and the suitability of each handling approach. The Ranker applies a scoring model to prioritize handling strategies that provide the most robust, context-specific solutions based on exception attributes and program impact. This graded feedback loop refines the recommendations by selecting strategies that minimize overhead and maximize reliability, as discussed in Section 4.3.
>
> *Adaptability and Continuous Refinement through Feedback Loops*
>
> The Handler agent also incorporates feedback from the environment, specifically during few-shot verification phases in Deep-RAG processing, which allows it to dynamically adjust and refine handling logic based on real-world performance. This feedback mechanism ensures that the handling strategies are not only theoretically sound but are also effective in practical scenarios, meeting both correctness and optimality standards.

---

### Official Review · Reviewer_fDWJ · 2024-11-05

**Soundness:** 1
**Presentation:** 1
**Contribution:** 1
**Rating:** 1
**Confidence:** 4

**Summary:**

This paper introduces Seeker, a tool for generating exception handling code with LLMs. Seeker is based on a multi-agent system that first locates the code location that needs exception handling, then performs RAG on a dataset of existing exception handling code, and finally generates the appropriate exception handling code. Seeker contributes to the standardization, interpretability, and generalization of automated exception handling generation.

**Strengths:**

- Some preliminary studies seems to be conducted to guide the development of Seeker.

**Weaknesses:**

- Paper writing is quite bad, to be honest, looks like automatically generated without much polishing. I have very hard time to follow the main contributions of this paper. Many terms are mentioned without proper explanation.

- The contribution is more on the engineering side than research. Namely Seeker is a combination of existing techniques: agent framework, RAG, few-shot learning, etc.

- Experiments were performed on a very limited number of open-source repositories, limiting the validity of the results.

**Questions:**

- lines 75-86: I cannot follow the concern about the "standardization" of existing exception handling techniques. What does it mean exactly, can you give an example?

- How did you come up with the "chain-of-thought used by senior human developers" in Figure 1(b), is it from user studies / interviews, or from any prior work? Also, it is hard to tell any valuable information from the figure, which is basically putting together a random set of buzz words.

- line 155-161: Can you specify how many codebases (repositories), code reviews, and exception handling examples have you studied in the preliminary study? What does "implementation results" mean?

- line 162-170: When you talk about these phenomena, specifically when mentioning things like "prompts without effective guidance information" "increasing the interpretability" "increasing the generalization information", can you specify which prompts you are referring to or comparing between? It is hard to map the four kinds of prompts with your text.

- The four kinds of prompts seem to be a key part of the technique. Looking at their definitions in Section 3.2 and the examples in figures 4 and 5, where do you get the specific exception types, code-level scenario analysis, and the handling strategy, for the Fine-grained {Reminding, Inspiring, Guiding} prompts, respectively? From the examples it looks like they are generated based on existing exception handling code snippets, but what happens when we apply your technique to new codebases without any existing exception handling code snippets?

- line 264-266: these are not three ways of "handle" exceptions. The throw keyword in the method signature is to declare a possible exception, the throw keyword in method body is to actually throw an exception, and only the try-catch block is for handling an exception.

- The steps in Seeker is not consistent. In Section 1 it was described as "Scanner, Detector, Predator, Ranker, and Handler", but in Section 3.3 it changes to "Planner, Detector, Predator, Ranker, and Handler".

- You are introducing several techniques/algorithms, namely "Common Exception Enumeration (CEE)" and "Deep Retrieval-Augmented Generation (Deep-RAG)", very superfically in the main text, and immediately redirect to the appendix. Please don't do that---the main text should be self-contained, so at least include some basic details and examples to explain how they work. I did check the part of appendix talking about CEE and Deep-RAG, but my impression is that they're just engineering contributions hidden behind the fancy word---CEE = few-shot learning, and Deep-RAG = a for loop doing RAG multiple times. Please correct me if I'm wrong.

- line 433: what is the CodeReviewModel you are using here to compute Automated Code Review Score?

- line 482: how is Code Review Score (6th metric) different from the Automated Code Review Score (1st metric)?

- line 486-490: In your experiment dataset, is there any overlapping between your evaluation examples and the examples used for RAG?

---

> ### Author Response · Authors · 2024-11-16
> **Strong rebuttal to Reviewer fDWJ**
>
> **Thank you very much for your careful and enthusiastic comments. We will respond to all the questions you raised in your comments and provide easy-to-understand explanations of the issues in combination with the revised version. Considering that some misunderstandings may come from the discussion in the first draft of the paper, or from your lack of understanding of the paper and related issues, we hope that you will fully discuss the rebuttal and the revised version information, give us your advice, and reconsider your score, based on our full confidence in this paper.**
>
> **Strengths:** Some preliminary studies seems to be conducted to guide the development of Seeker.
>
> **Weakness1:** I have very hard time to follow the main contributions of this paper.
>
> **Response:** Thank you for the comment. We try to more clearly indicate contributions in the rebuttal revision, including:
>
> 1. We highlight the importance of standardization, interpretability, and generalizability in exception handling mechanisms, identifying a gap in existing research.
>
> 2. We propose Seeker, which decomposes exception handling into specialized tasks and incorporates Common Exception Enumeration (CEE) to enhance performance.
>
> 3. We introduce a deep retrieval-augmented generation (Deep-RAG) algorithm tailored for complex inheritance relationships, improving retrieval efficiency.
>
> 4. We conduct extensive experiments demonstrating that Seeker improves code robustness and exception handling performance in LLM-generated code.
>
> **Weakness2:** Paper writing is quite bad, to be honest, looks like automatically generated without much polishing.
>
> **Response:** Thank you for pointing out the writing problems. First, I would like to explain that the word "automatically generated" is not used anywhere in this paper. Without considering the comments of other reviewers——"The paper is well-written and easy to follow.", we hope that you can give us instructions on the specific content that needs to be optimized. We will explain the writing logic of the first draft and the revised version to you separately, and briefly summarize the main points of the content for your understanding.
>
> **Weakness3:** Many terms are mentioned without proper explanation.
>
> **Response:** Thank you for your comments on the readability of the article. What specific terms did you not understand? I think we can provide good explanations for any terms that appear in the article and that you find confusing. In conjunction with the revised version, we believe that there are no "innovative" or "extremely complex" terms for experts in the field. If you think there are still some, we can discuss them in depth at the rebuttal stage.
>
> **Weakness4:** The contribution is more on the engineering side than research. Namely Seeker is a combination of existing techniques: agent framework, RAG, few-shot learning, etc.
>
> **Response:** First of all, thank you for your recognition of the engineering contribution of the article. We generally believe that existing scientific research does not only include the underlying innovation of methods. In addition to data and algorithms, it is also an important contribution to combine existing algorithms and make acceptable breakthroughs on a long-standing problem that has no good solution. Especially for the field of AI, all research seems to be the engineering migration of transformers. We believe that in this article, we not only tried to solve the long-standing difficult problem of exception handling in software engineering (from Westley Weimer and George C. Necula. Finding and preventing run-time error handling mistakes. In OOPSLA, 2004. to Yuchen Cai, Aashish Yadavally, Abhishek Mishra, Genesis Montejo, and Tien Nguyen. Programming Assistant for Exception Handling with CodeBERT. In ICSE, 2024 until now, this article), but also provided an acceptable solution. Academically, we also pointed out the activation Agent Framework and Deep-RAG methods in the face of complex inheritance relationships, and proposed a more effective few-shot learning solution in processing the sample data and granularity of the retrieved documents. We believe that each step is not a simple call to traditional or original work, but rather a valuable optimization method for specific problems or even more generalized features.
>
> **Weakness5:** Experiments were performed on a very limited number of open-source repositories, limiting the validity of the results.
>
> **Response:** Thank you very much for your comments on the experimental settings. Even though our test set must rely on open source projects: because they are real, high-complexity development scenarios where actual exception handling is carried out, we still made targeted data adjustments on them. (Please see next block due to the constraint of context window.)

---

> ### Author Response · Authors · 2024-11-16
> **Strong rebuttal to Reviewer fDWJ (Part 2)**
>
> (Continue Comment) For example, to prevent data leakage, we made regular modifications to the small-scale dataset to verify that our test results are stable against data leakage issues; in the revised Appendix Experiment Figure 7, we added more experiments to demonstrate that we are stable in terms of project creation time and project domain compared to other baselines. In addition, we also ensured the sufficiency of the experimental data, including 750 high-quality test samples, which is larger than the common code evaluation benchmarks and parallel test sets. If we do not conduct experiments with this "limited number of open-source repositories", do you have any better suggestions? Because this is very inconsistent with the common research methods in the field of software engineering. For example, manually writing projects? This will limit the complexity and authenticity of the actual test code. The experiments on KPC proved that the disconnection from the real project is unacceptable. What kind of experimental settings do you suggest to achieve the "validity of the results" you specifically refer to?
>
> **Question1:** lines 75-86: I cannot follow the concern about the "standardization" of existing exception handling techniques. What does it mean exactly, can you give an example?
>
> **Response:** We appreciate the reviewer’s request for clarification regarding the "standardization" of exception handling practices and the challenges this poses. The term "standardization" in this context refers to the absence of widely accepted and consistently applied norms or quality standards for writing and implementing exception handling code. Currently, developers often have substantial flexibility in how they choose to handle exceptions, leading to a variety of practices that vary widely in quality and effectiveness. This inconsistency is problematic because certain “bad practices,” such as exception swallowing (where exceptions are caught but not properly handled) and overly generic exception handling (e.g., using default catch-all exceptions), lead to fragile code structures that are prone to errors and bugs.
>
> For example, the phenomenon of "swallowing" exceptions, where exceptions are caught without any meaningful error management, illustrates this issue. Such practices not only obscure the root causes of failures but also make debugging and maintenance significantly more challenging. Another common example involves developers using overly broad exception handlers, which catch many unrelated exceptions indiscriminately, rather than capturing specific exceptions that provide valuable context. These practices lead to code that is difficult to maintain and lacks robustness. As noted in relevant research, such as the study on Exception Handling Bugs, these issues are among the most frequent sources of errors in software projects, underscoring the urgent need for standardized exception handling approaches to improve code reliability.
>
> By establishing a standardized approach, including defining best practices and quality benchmarks, developers could more consistently produce robust and maintainable code. We discuss common misunderstandings about exception handling in detail in Section 3.1 of our paper.
>
> **Question2:** How did you come up with the "chain-of-thought used by senior human developers" in Figure 1(b), is it from user studies / interviews, or from any prior work? Also, it is hard to tell any valuable information from the figure, which is basically putting together a random set of buzz words.
>
> **Response:** Thank you for your question regarding the development of the "chain-of-thought" illustrated in Figure 1(b). This framework synthesizes concepts from both industry insights and prior research to depict the thought process of senior developers when handling exceptions. The structure of the chain-of-thought model was informed by:
>
> 1. Prior Research in Exception Handling:
> Exception Range and Sensitive Code: Prior work, such as Nakshatri et al. (2016) and de Pádua & Shang (2017), emphasizes the importance of identifying specific exception types and capturing lower-level exceptions within the class hierarchy for improved debugging and code robustness. These studies indicate that developers benefit from targeting specific exception types rather than overly broad handlers, which supports the inclusion of “Exception Range” and “Sensitive Code” as core components in the model.
>
> 2. Insights from Industry Consultation:
> To complement the research literature, we consulted with experienced engineers from industrial company, who provided insights into the practical aspects of exception handling. The expert emphasized the importance of drawing on programming experience and specific knowledge resources (like SDK documentation) to manage complex exception handling cases. This feedback informed the inclusion of “Programming Experience” and “Knowledge Invocation” as part of the developer thought process.

---

> ### Author Response · Authors · 2024-11-16
> **Strong rebuttal to Reviewer fDWJ (Part 3)**
>
> (Continue Comment)
>
> 3. Key Terms in Our Experimental Setup:
> Each term in the figure directly links to elements of our experiments and module design. Exception Matching and Handling Design with cues from our Common Exception Enumeration (CEE) document is our basic logic of generalized information. “Grammar Mastery” and “SDK Learning” help reinforce accurate exception targeting, while “Handling Method” and “Resolution Effects” provide structured guidance on selecting and evaluating exception handling strategies.
>
> 4. Clarification on the Model's Purpose:
> While these elements may appear as standalone terms, they represent structured stages in a senior developer’s approach to handling exceptions, as supported by empirical studies and industry practices. We can enhance Figure 1(b) to make these relationships clearer, showing how each component fits into a systematic workflow rather than a list of separate “buzzwords.”
>
> **Question3:** line 155-161: Can you specify how many codebases (repositories), code reviews, and exception handling examples have you studied in the preliminary study? What does "implementation results" mean?
>
> **Response:** Thank you for your question. In the preliminary study, we analyzed 100 code snippets sourced from a total of 20 distinct repositories to ensure a representative sample of real-world code. Additionally, we reviewed multiple research papers from academic conferences and official programming language documentation, which provided valuable insights into common exception handling practices and challenges. These resources helped us establish the context and conclusions presented in our study.
>
> Regarding “implementation results,” this phrase refers to the specific outcomes and observations gathered during the workflow of our preliminary study. The term is a phrase indicating the results derived from systematically implementing and testing the designed prompts across the selected code snippets. This includes insights on the influence of varying prompt levels on exception handling quality, as detailed in our methodology.
>
> **Question4:** line 162-170: When you talk about these phenomena, specifically when mentioning things like "prompts without effective guidance information" "increasing the interpretability" "increasing the generalization information", can you specify which prompts you are referring to or comparing between? It is hard to map the four kinds of prompts with your text.
>
> **Response:** Thank you for your question regarding the specific prompts and the distinction among different types. To clarify:
>
> 1. Reference to Specific Prompts: In our study, we compare four types of prompts—General prompting, Coarse-grained Knowledge-driven prompting,  Fine-grained Knowledge-driven prompting and Fine-grained Knowledge-driven with handling logic prompting—each representing a progressive enhancement in the quality of guidance provided to developers or models. These prompts are sequentially designed to reveal different aspects of exception handling by gradually incorporating interpretability and generalization information. For ease of reference, we presented the finalized version of these prompts in the Appendix, which shows the distinct levels of information across each type.
>
> 2. Three Key Characteristics in Prompts:
>
> Effective Guidance Information: “Guidance information” refers to the structured and progressive information embedded in our prompts, which increases across the four types. For instance, Coarse-grained Reminding includes basic reminders about exceptions but lacks specific cues, while Fine-grained Knowledge-driven with handling logic prompting adds in-depth context and actionable exception handling strategies. This tiered approach allows us to study how different levels of guidance impact exception handling performance.
>
> Interpretability: Interpretability within our prompts is most prominently featured in the General prompting and Knowledge-driven prompting. This characteristic involves providing developers with clear, scenario-based explanations and contextual cues, which help clarify the code’s potential vulnerabilities. In particular, these prompts use detailed exception scenarios derived from our CEE (Common Exception Enumeration) nodes, enabling developers to better understand the specific context and risk of fragile code areas.
>
> Generalization Information: Generalization information is primarily represented in our Fine-grained Guiding prompt, where we include best practices and standardized handling strategies sourced from our CEE document nodes. The CEE includes comprehensive exception type guidelines and handling norms, giving developers a structured approach to managing both common and long-tail exceptions. The generalization information ensures that exception handling solutions are not just case-specific but adhere to a broader, consistent standard, making code more robust and maintainable.

---

> ### Author Response · Authors · 2024-11-16
> **Strong rebuttal to Reviewer fDWJ (Part 4)**
>
> (Continue Comment) If additional detail is needed, we would be happy to further elaborate on these prompt types and characteristics.
>
> **Question5:** The four kinds of prompts seem to be a key part of the technique. Looking at their definitions in Section 3.2 and the examples in figures 4 and 5, where do you get the specific exception types, code-level scenario analysis, and the handling strategy, for the Fine-grained {Reminding, Inspiring, Guiding} prompts, respectively? From the examples it looks like they are generated based on existing exception handling code snippets, but what happens when we apply your technique to new codebases without any existing exception handling code snippets?
>
> **Response:** Thank you for your insightful question. We understand the importance of clarifying the prompt generation mechanisms, especially for new codebases without existing exception handling references. We believe that Figures 1-5, as well as our methodology section (particularly the description of our multi-agent approach in Section 3.3), comprehensively outline the flexibility of prompt content and the adaptability of our algorithm across codebases. Here, we provide further clarification.
>
> 1. Prompt Content Derivation and Flexibility:
>
> The Fine-grained {Reminding, Inspiring, Guiding} prompts in our methodology are indeed designed to capture specific exception types, code-level scenarios, and handling strategies. As detailed in Section 3.2 of the revised paper, these prompts are generated by an adaptable agent-driven mechanism where each agent gathers context-specific information.
>
> The fine-tuned prompts do not depend solely on existing exception handling code snippets. Instead, they derive exception-specific scenarios and strategies using our Common Exception Enumeration (CEE) and the Deep-RAG model, which references real-world handling best practices and comprehensive exception documentation. This ensures our technique’s applicability even when no pre-existing code-specific exception handling is available in the new codebases.
>
> 2. Adaptability for New Codebases:
>
> When applying Seeker to a new codebase, the prompts are dynamically generated based on the code slice context rather than static code snippets. As outlined in the Methodology (Section 3.3), each agent—particularly the Detector and Predator agents—conducts scenario and property matching with both general and specific exception handling requirements, thereby identifying relevant exception types even in new environments.
>
> The CEE serves as a repository of best practices, offering scenario descriptions, handling strategies, and exceptions that are adaptable to varied codebases without the need for specific historical exception handling code. The adaptability of Seeker to dynamically interpret the code context, rather than relying on existing code snippets, is crucial for its performance in unfamiliar codebases.
>
> 3. Algorithm Process and Prompt Flexibility:
>
> As illustrated in the workflow diagram (Figure 3), each agent captures information specifically required by the portion of the code it interacts with, ensuring that each prompt precisely matches the exception handling needs of the current code segment.
>
> The prompts follow a progressive structure (Reminding, Inspiring, Guiding) to enhance flexibility in their application, as shown in Figure 1. This ensures that Seeker adapts the level of guidance provided based on the exception types and code scenarios identified. Thus, prompt flexibility is inherent in the multi-agent system, which tailors exception handling strategies based on both fine-grained interpretability and generalizability for new or modified codebases.
>
> We hope this explanation addresses your concerns regarding the adaptability and flexibility of our prompts across different codebases. Thank you for helping us enhance the clarity of our approach.
>
> **Question6:** line 264-266: these are not three ways of "handle" exceptions. The throw keyword in the method signature is to declare a possible exception, the throw keyword in method body is to actually throw an exception, and only the try-catch block is for handling an exception.
>
> **Response:** Clarification on Exception Handling Terminology and Practice:
>
> In lines 264-266 of the original submission, our intention was to outline the conventional ways of approaching exception-related scenarios in code (i.e., declaration, propagation, and handling). The structure we used was aimed at providing a simplified framework for discussing exception mechanisms in the context of our multi-agent model, Seeker, which focuses on refining these processes through automated handling. We recognize, as you noted, that only the try-catch block is primarily used for handling exceptions, while the throw and throws keywords respectively serve different purposes: throwing exceptions and declaring possible exceptions in method signatures.

---

> ### Author Response · Authors · 2024-11-16
> **Strong rebuttal to Reviewer fDWJ (Part 5)**
>
> (Continue Comment) Alignment with Reviewer’s Expertise and Support of Research Objective:
>
> The distinction you raised aligns well with the real-world practices that professional developers, like yourself, recognize. This very distinction supports the core objective of our work, which is to address the nuances in exception handling by aiding large language models (LLMs) with structured guidance on detection, capture, and handling. In our method, we prioritize the try-catch block as the preferred handling mechanism, as indicated by industry best practices, while acknowledging the other uses of exception declarations.
>
> **Question7:** The steps in Seeker is not consistent. In Section 1 it was described as "Scanner, Detector, Predator, Ranker, and Handler", but in Section 3.3 it changes to "Planner, Detector, Predator, Ranker, and Handler".
>
> **Response:** Thank you for identifying this issue regarding the naming of Seeker’s components in different sections of our initial submission. We acknowledge the inconsistency in the terminology used for Seeker's agent structure, which may have caused confusion.In our revised manuscript, we have standardized the component names of Seeker across all relevant sections. We have ensured consistent use of the terms “Planner, Detector, Predator, Ranker, and Handler” throughout the paper. This inconsistency arose from updates made during the revision phase, where we refined Seeker’s agent flow to better align with the system's intended functionality. The naming discrepancy was an unintended artifact from an earlier draft. We appreciate your attention to this detail, which helped us improve the clarity of the manuscript.
>
> **Question8:** You are introducing several techniques/algorithms, namely "Common Exception Enumeration (CEE)" and "Deep Retrieval-Augmented Generation (Deep-RAG)", very superfically in the main text, and immediately redirect to the appendix.
>
> Please don't do that---the main text should be self-contained, so at least include some basic details and examples to explain how they work.
>
> I did check the part of appendix talking about CEE and Deep-RAG, but my impression is that they're just engineering contributions hidden behind the fancy word---CEE = few-shot learning, and Deep-RAG = a for loop doing RAG multiple times. Please correct me if I'm wrong.
>
> **Response:** We appreciate your feedback and the opportunity to clarify the contributions and implementations of CEE and Deep-RAG. Below, we address each of your points:
>
> 1. Main Text Structure and Content Decisions:
>
> We understand the importance of making the main text self-contained. In the initial submission, due to page limitations, we prioritized content on motivation, the broader framework, and the overall experimental impact, deferring some algorithmic details to the appendix. However, based on your feedback, we will revise the main text to include essential explanations and examples of CEE and Deep-RAG to ensure a clearer, more integrated understanding.
>
> Contextual Justification for the Appendix: Given the complexity of our method, specific algorithmic details require additional space to avoid oversimplification. The appendix provides extended technical elaboration and decision-making insights, which we view as valuable for those interested in replicating or extending our approach. Our revised structure will provide readers with a self-contained overview while preserving the appendix's detailed role.
>
> 2. Clarifying the Role and Novelty of Common Exception Enumeration (CEE):
>
> Clarification: Contrary to the impression that CEE is solely an application of few-shot learning, it is a carefully structured knowledge base and standardization effort for exception handling scenarios. CEE builds on a Java exception hierarchy, similar in structure to CWE (Common Weakness Enumeration), but specifically tailored to exception handling and robustness in code generation by LLMs.
>
> Contribution and Value: CEE includes comprehensive exception scenarios, properties, and handling logic. By providing these contextually enriched descriptions, CEE serves as a reusable, community-contributable resource designed to improve exception handling practices, supporting developers and LLMs in systematically identifying and managing exceptions. We will clarify this distinction in the main text and provide examples of CEE entries to showcase the specific details and handling strategies.

---

> ### Author Response · Authors · 2024-11-16
> **Strong rebuttal to Reviewer fDWJ (Part 6)**
>
> (Continue Comment) 3. Deep Retrieval-Augmented Generation (Deep-RAG) as a Novel Method Beyond Multiple Iterative Retrievals:
>
> Clarification: Deep-RAG is more than a simple loop for RAG processes. It introduces a multi-layered, scenario-labeled exception tree that enables hierarchical and context-specific exception handling. The Deep-RAG structure leverages inheritance relationships within exception classes to improve specificity in exception handling, incorporating depth-wise evaluations of nodes that account for domain context and program fragility.
>
> Detailed Contribution: Our approach goes beyond traditional retrieval processes by adding feedback-driven refinement at each level, which enhances accuracy. This process is both computationally optimized and ensures that RAG’s retrieval performance is scaled effectively, with parallel processing of exception nodes to improve precision in complex exception hierarchies. We will emphasize this mechanism and its contributions more thoroughly in the revised main text.
>
> 4. Reviewer’s Interpretation of CEE and Deep-RAG as ‘Fancy Words’ for Basic Techniques:
>
> We appreciate your interpretation, but as highlighted, CEE and Deep-RAG are structured methodologies that offer a novel approach to improving exception handling. These are carefully designed to address specific limitations in the traditional application of RAG and few-shot learning by introducing enhanced specificity and performance in exception handling contexts. We will provide further elaboration on this in the main text with examples to better illustrate the innovation and problem-solving capacity embedded in each technique.
>
> In summary, we acknowledge the feedback regarding the presentation of CEE and Deep-RAG and will adjust our manuscript to offer more clarity and detail within the main text, maintaining the appendix for comprehensive technical elaboration. Thank you for your constructive comments, which will help enhance the accessibility and rigor of our work.
>
> **Question9:** line 433: what is the CodeReviewModel you are using here to compute Automated Code Review Score?
>
> line 482: how is Code Review Score (6th metric) different from the Automated Code Review Score (1st metric)?
>
> **Response:** *Explanation of the CodeReviewModel used to compute Automated Code Review Score (Line 433)*
>
> In our paper, the Automated Code Review Score (ACRS) is computed using an automated code review tool that checks each code segment for adherence to predefined coding standards and best practices. In our experiment, we use Code Llama-34B. Specifically, ACRS is calculated as follows:
>
> $$
> \text{ACRS} = \frac{\sum_{i=1}^{N} w_i s_i}{\sum_{i=1}^{N} w_i} \times 100\%
> $$
>
>
> This model evaluates the structural quality of the generated code by checking adherence to best practices and standards. A higher ACRS score signifies a higher degree of compliance with quality standards.
>
> *Difference between Code Review Score (CRS) and Automated Code Review Score (ACRS) (Line 482)*
>
> The difference between Code Review Score (CRS) and Automated Code Review Score (ACRS) can be summarized as follows:
>
> Automated Code Review Score (ACRS): This score is derived from an automated tool that evaluates adherence to specific coding rules and quality standards, providing a quantitative assessment based on a weighted formula, as defined above. It primarily reflects the objective compliance of the generated code with coding standards.
>
> Code Review Score (CRS): This score is computed by submitting the generated try-catch blocks to a large language model-based reviewer (e.g., GPT-4o), which provides a binary assessment (good or bad) of each block. The CRS formula is:
>
> $$
> \text{CRS} = \frac{N_{\text{good}}}{N_{\text{total}}} \times 100\%
> $$
>
> CRS reflects the proportion of code blocks that meet engineering best practices, as assessed by the language model, providing an additional qualitative layer to the automated review.
>
> **Question10:** line 486-490: In your experiment dataset, is there any overlapping between your evaluation examples and the examples used for RAG?
>
> **Response:** *Clarification of CEE Dataset Composition*
>
> The Common Exception Enumeration (CEE) was specifically constructed to improve exception handling guidance and was not derived from any part of the experimental dataset used in evaluation. As stated in our revised paper, CEE’s foundation primarily involves two elements: the official documentation of exceptions from the Java Development Kit (JDK) and comprehensive coding standards from high-quality code review practices. This ensures the separation between the development of CEE and our experimental evaluation dataset.

---

> ### Author Response · Authors · 2024-11-16
> **Strong rebuttal to Reviewer fDWJ (Part 7)**
>
> (Continue Comment) *Derivation Process and Data Integrity*
>
> As described in Section 3.2 of our paper, the CEE was created through a meticulous process focusing on real-world Java exception handling practices, taking input from external resources without reference to specific examples in our evaluation set. We ensured that high-quality code standards and external resources, such as exception documentation and developer guidelines, were employed without any direct or indirect incorporation of evaluation data samples.
>
> *Evidence of Non-Overlap in Experiment Setup*
>
> In the revised paper, we have elaborated on the construction methodology of CEE (Section 3.2 and Appendix A.1.2). This methodological distinction supports the absence of overlap, as CEE’s purpose is to offer generalized handling rules rather than being derived from evaluation-specific data.
>
> **We have carefully read the opinions and scores of the three reviewers. I personally think that the problems they pointed out come from their lack of understanding of the field itself (such as confidence 2) (and some lack of common sense in software engineering). On the other hand, as a mechanism that is not so well-known/skilled, many reviewers themselves do not understand the importance and difficulty of exception handling, while the fact is that it is very important in actual development (due to the double-blind policy, I think I cannot list Internet companies with such urgent needs). Therefore, it is understandable that the reviewers raised some confusions that have been explained as much as possible in the article, and they can try to explain them clearly. We hope that you (here refers to PC, reviewers) can comprehensively consider the motivation, method details, and experimental results, and give objective review results.**

---

> > ### Comment · Reviewer_fDWJ · 2024-11-16
> > **Is there a human author on this paper???**
> >
> > Thank you for writing (or should I say, generating) the detailed response. Despite the fact that it is written in fluent English and is extraordinary long, the response is mostly not understandable. Anyway, please see below my responses to your answers to my questions.
> >
> > > We have carefully read the opinions and scores of the three reviewers. I personally think that the problems they pointed out come from their lack of understanding of the field itself (such as confidence 2) (and some lack of common sense in software engineering). On the other hand, as a mechanism that is not so well-known/skilled, many reviewers themselves do not understand the importance and difficulty of exception handling, while the fact is that it is very important in actual development (due to the double-blind policy, I think I cannot list Internet companies with such urgent needs). Therefore, it is understandable that the reviewers raised some confusions that have been explained as much as possible in the article, and they can try to explain them clearly. We hope that you (here refers to PC, reviewers) can comprehensively consider the motivation, method details, and experimental results, and give objective review results.
> >
> > Please, respect your reviewers and their time. We are assigned to this paper because we have expertise in software engineering, know what is exception handling (I use it in coding everyday), and most importantly, can tell if a paper is written by human or GenAI. Maybe you have some great research done, but that can never be delivered by putting together LLM-generated text without basic checking for consistency.
> >
> > Re Re Question 1: According to your explanation, "standardization" means that all developers should follow the same convention for writing exception handling code. How is Seeker supporting this goal? I didn't see anything in the later sections of the paper that explains how Seeker integrates standardization into its technique or measure standardization in the evaluation.
> >
> > Re Re Question 2: These are great answers and should all be included in the paper; in the original manuscript, there was no explanation of Figure 1(b) in text.
> >
> > Re Re Question 3: If you look at your manuscript, Figure 1(a) is a line plot (with an unclear caption "Our preliminary tendency") and you're calling it the "implementation results". I am asking what exact are the results you are implementing here and what numbers you are using.
> >
> > Re Re Question 4: Ok, in your answer you just listed me the four prompts and three characteristics again. What are the mappings between them?
> >
> > Re Re Question 5: Thank you for pointing out that there is a Figure 3 in the paper. I considered figures with reference count of 0 as not being used in the paper. It seems that the CEE database is extracted from the same set of repositories as the evaluation dataset, so finding a closely-related exception handling code snippet is not difficult. Have you evaluated Seeker on a repository that is not used when creating CEE?
> >
> > Re Re Question 6: Please update the terminology you use in the paper. I didn't get what your second part of the answer is trying to say.
> >
> > Re Re Question 7: Thanks for fixing this.
> >
> > Re Re Question 8: Just as you described, the natural of CEE and Deep-RAG are few-shot learning and RAG, respectively. Of course, you will need to collect a dataset of few-shot samples and design a retrieval algorithm corresponding to the application domain, which is generating exception handling code in your case. I don't see any novel research contribution here.
> >
> > Re Re Question 9: Can you get a human author to read your answer and explain to me: what is the difference between the two metrics?
> >
> > Re Re Question 10: Is there an overlap in the set of repositories that you use in the evaluation and use as the source of CEE?

---

> > > ### Author Response · Authors · 2024-11-16
> > > **1:)  <= This must be an AI response**
> > >
> > > First of all, we respect all reviewers. As a reviewer, I know how it feels to review a bad article: but not this one. We carefully analyzed and thanked you for your question before answering each question. I think this is out of respect rather than perfunctory. You mentioned again that except for this article, even the replies are automatically generated. First of all, I admire your ability to identify AI. Based on your outstanding observation, I put forward such a hypothesis: if the author of the article is not a native speaker (but you are), Google Translate will be used to convert the language when dealing with some complex semantics and then manually reviewed. Maybe we think this is the correct semantics, but it is incoherent in your opinion and looks like generated text? We hope you can explain the rationality of this place and whether your own judgment may confuse the difference between AI generation and AI translation.
> > >
> > > As you said, you have rich experience in software engineering and knowledge of exception handling mechanisms. Okay, we believe this for the time being, because the "standardization" you repeatedly questioned is something that a developer with rich development experience should be able to easily recognize-no other reviewer has raised the same question. Here I think we did a very good job in the response to Part 2. We showed you a full picture of "standardization" from the normative definition and the non-normative special cases. This is a processing specification from the demand mapping to the specific strategy. In the actual work of Seeker, it is reflected in Seeker's ability to detect vulnerable code (specific lines) more accurately/normatively without causing false positives or missed negatives, to capture exception types more fine-grained (specific)/normatively without false capture or coarse-grained capture (we have cited and discussed the hazards of this non-normativeness in detail in Section 3.1), and to select exception handling strategies that are more in line with the requirements. Do you get the meaning of "standardization"? The core point of our work is to solve "standardization". In fact, our various sub-task indicators illustrate the improvement of "standardization". Let me give you another example to help you understand. I think since you are also doing research on AI4SE, you must know the fallback mechanism in the model API call, which is the most typical engineering exception handling. In the complex algorithm framework of Multi-Agent (just like Seeker), if each API is not accurately maintained, an error in the algorithm will enter the exception branch and cause the program to function incorrectly. Therefore, the fallback mechanism must accurately locate the model API call code line; if we choose a coarse-grained exception type, then whether it is a problem with the model API itself or the network fluctuation problem we intend to solve, it will cause fallback to occupy resources; finally, the template and parameter adjustment of the fallback mechanism are used by us as a strategy for benchmarking requirements. You must be very familiar with this example? Can you understand it in combination with your daily development?
> > >
> > > Re Re Re Question1: As we said in the example above, each Agent part of Seeker intends to solve the "standardization" problem. Detector is used to locate vulnerable code from manageable units at a coarse granularity, Predator uses specific flags to enter code line selection in the vulnerable code block, and Ranker is used to comprehensively handle strategies and exception types on the same inheritance branch to select the most reasonable exception handling method. Where is your misunderstanding? The indicators of ACRS and CRS correspond to the overall exception handling quality, COV corresponds to the performance of Detector, COV-P corresponds to the performance of Predator selection, ACC corresponds to the performance of Ranker's exception type selection, and ES corresponds to the correlation of exception handling strategies compared with golden. What do you not understand? Do we need to express these obvious confusions in our replies to be more "human"?
> > >
> > > Re Re Re Question 3: I think our reply accurately locates the part you are referring to and explains it very clearly. Fig1 (a) appears at the beginning because logically this figure is a pre-experimental relationship diagram, which is used to combine KPC to assist in demonstrating the effective information of exception handling for LLM/Human. The indicators come from the code review of the test code block. The experimental settings are very clear in Section 2.1 of the two versions of the original text. We don't understand what you are confused about?

---

> > > > ### Author Response · Authors · 2024-11-16
> > > > **2:) <= This must be an AI response**
> > > >
> > > > Re Re Re Question 4: I think the source of this question is simply because you didn't see the example fig5 we put in the appendix and therefore couldn't understand the meaning of these terms? Indeed, based on the fact that you haven't read the KPC article and don't understand "standardization", we should indeed be aware of this and give you a specific example. Moreover, we have also clearly shown the information transmission of Agents Prompt, CEE Sample and Agent Framework in fig3. I think it is impossible to ask such a low-level question if you don't have a preconceived concept about this article? How else do we need to explain, without looking at the example? Because you seem to be unable to understand it anyway.
> > > >
> > > > Re Re Re Question 5: Yes, but it is obvious that this "Seeker Work Flow" is the most important overview diagram in this article. We are surprised that you read and index the entire paper completely according to the text. This is a difference in reading methods, and I reserve my opinion. But after you saw this picture, didn't you think about the previous question at all? Because there is also a clear example of the process in this picture, doesn't it help your understanding?
> > > >
> > > > Of course, please read the experiment and our response carefully. In the appendix experiment, we even evaluated the variant test set, CoderEval and SWE-bench (Java), which are not in our CEE build library.
> > > >
> > > > The analysis of the original reply is good if you already understand the experimental flow: The fine-tuned prompts do not depend solely on existing exception handling code snippets. Instead, they derive exception-specific scenarios and strategies using our Common Exception Enumeration (CEE) and the Deep-RAG model, which references real-world handling best practices and comprehensive exception documentation. This ensures our technique’s applicability even when no pre-existing code-specific exception handling is available in the new codebases. When applying Seeker to a new codebase, the prompts are dynamically generated based on the code slice context rather than static code snippets. As outlined in the Methodology (Section 3.3), each agent—particularly the Detector and Predator agents—conducts scenario and property matching with both general and specific exception handling requirements, thereby identifying relevant exception types even in new environments. The CEE serves as a repository of best practices, scenario offering descriptions, handling strategies, and exceptions that are adaptable to varied codebases without the need for specific historical exception handling code. The adaptability of Seeker to dynamically interpret the code context, rather than relying on existing code snippets, is crucial for its performance in unfamiliar codebases.
> > > >
> > > > Re Re Re Question 6: To put it simply, your point of view and the fact that only try-catch belongs to exception handling do not conflict with the discussion of our work. Based on previous work, we are sure that there is no problem with the statement here, and it is a complete explanation for reviewers who may have relevant doubts.
> > > >
> > > > Re Re Question 8: Interpretation from the original reply again——Contrary to the impression that CEE is solely an application of few-shot learning, it is a carefully structured knowledge base and standardization effort for exception handling scenarios. CEE builds on a Java exception hierarchy, similar in structure to CWE (Common Weakness Enumeration), but specifically tailored to exception handling and robustness in code generation by LLMs.
> > > >
> > > > First, the main contribution of CEE is not reflected in the method: CEE serves as a reusable, community-contributable resource designed to improve exception handling practices, supporting developers and LLMs in systematically identifying and managing exceptions
> > > >
> > > > If you think that such standardized documents are worthless, then CWE and various Empirical Studies should be meaningless. We summarize these important experiences so that human/LLM developers can deal with exception handling problems more effectively.
> > > >
> > > > Second, for the complex inheritance relationship of Exception, the construction of each label is not just the application of "few-shot learning", over. Exceptions are closely interdependent. In many cases, their scenarios and activation details overlap. The global unification of granularity and the final quality assurance have deviated from the main line of "few-shot learning". Let me repeat the question. If you think that engineering tuning is not innovation, then what is the difference between any of the work we are doing now and the engineering tuning of transformers? Can you answer?

---

> ### Author Response · Authors · 2024-11-16
> **3:) <= This must be an AI response**
>
> Let's talk about RAG. I think your understanding of RAG is very worrying. One sentence that is essentially based on RAG can deny Graph-RAG and Self-RAG, and the field of information retrieval can go bankrupt. In fact, Deep-RAG can be generalized to solve very important general problems. As a senior software engineering scholar, you must know that inheritance, long texts and complex dependencies are the three main problems that existing research has encountered difficulties. Now that you have seen the presentation of Deep-RAG in fig3 and the algorithm flow chart, you may be aware of the application potential of this algorithm in the main problem of inheritance relationship - perhaps you think it is easy to achieve accurate retrieval of similar text similarity with traditional RAG? Deep-RAG is more than a simple loop for RAG processes. It introduces a multi-layered, scenario-labeled exception tree that enables hierarchical and context-specific exception handling. The Deep-RAG structure leverages inheritance relationships within exception classes to improve specificity in exception handling, incorporating depth-wise evaluations of nodes that account for domain context and program fragility.
>
> Detailed Contribution: Our approach goes beyond traditional retrieval processes by adding feedback-driven refinement at each level, which enhances accuracy. This process is both computationally optimized and ensures that RAG’s retrieval performance is scaled effectively, with parallel processing of exception nodes to improve precision in complex hierarchies.
>
> **In a macro sense, Deep-RAG brings the activation relationship of neurons from the neural network operator level to the text level. Is this an innovation? **
>
> Re Re Re Question 9: I will use one sentence to show that you don’t understand software engineering evaluation at all: "Is the score of the code review model a binary classification?" We need to further explain to you the basic common sense. Is the basic quality score of the code review A-F?
>
> **Just as you have no reason to believe that all texts are AI-generated (no reviewer has ever slandered me with such an accusation), I would like to appeal again that your understanding of the entire article and even the entire field is very limited, and I question your professionalism. **
>
> Don't you understand where the new quantitative indicators are written?
>
> Re Re Re Question 10: Original reply-CEE was not derived from any part of the experimental dataset used in evaluation. As stated in our revised paper, CEE’s foundation primarily involves two elements: the official documentation of exceptions from the Java Development Kit (JDK) and comprehensive coding standards from high-quality code review practices. This ensures the separation between the development of CEE and our experimental evaluation dataset.
>
> Why do you always doubt this most basic overfitting problem of ML? Where did we show low scientific research literacy to give you this most basic misunderstanding? If we can find 100 high-quality and standardized test sets, why don't we divide them to use 50 code bases to assist in fine-tuning and build CEE samples, and 50 code bases for testing? Our original response should have explained it clearly, what is the point of repeating this question? Considering your comments on Question 5, it seems that you have some preconceived notions, I hope you can explain it.

---

### Note · Authors · 2025-01-23

I have read and agree with the venue's withdrawal policy on behalf of myself and my co-authors.